# Ub-ProT reveals global length and composition of protein ubiquitylation in cells

Hikaru Tsuchiya[1], Daocharad Burana[1,2], Fumiaki Ohtake[1], Naoko Arai[1], Ai Kaiho[1], Masayuki Komada[2], Keiji Tanaka[1] & Yasushi Saeki [1]

Protein ubiquitylation regulates diverse cellular processes via distinct ubiquitin chains that differ by linkage type and length. However, a comprehensive method for measuring these properties has not been developed. Here we describe a method for assessing the length of substrate-attached polyubiquitin chains, "ubiquitin chain protection from trypsinization (Ub-ProT)." Using Ub-ProT, we found that most ubiquitylated substrates in yeast-soluble lysate are attached to chains of up to seven ubiquitin molecules. Inactivation of the ubiquitin-selective chaperone Cdc48 caused a dramatic increase in chain lengths on substrate proteins, suggesting that Cdc48 complex terminates chain elongation by substrate extraction. In mammalian cells, we found that ligand-activated epidermal growth factor receptor (EGFR) is rapidly modified with K63-linked tetra- to hexa-ubiquitin chains following EGF treatment in human cells. Thus, the Ub-ProT method can contribute to our understanding of mechanisms regulating physiological ubiquitin chain lengths and composition.

[1] Laboratory of Protein Metabolism, Tokyo Metropolitan Institute of Medical Science, 2-1-6 Kamikitazawa, Setagaya-ku, Tokyo 156-8506, Japan. [2] Cell Biology Center, Institute of Innovative Research, Tokyo Institute of Technology, 4259-B16 Nagatsuta, Midori, Yokohama 226-8501, Japan. Hikaru Tsuchiya and Daocharad Burana contributed equally to this work. Correspondence and requests for materials should be addressed to K.T. (email: tanaka-kj@igakuken.or.jp) or to Y.S. (email: saeki-ys@igakuken.or.jp)

Protein ubiquitylation is a dynamic multifaceted post-translational modification (PTM) responsible for regulating a diverse array of cellular processes, including protein degradation, protein trafficking, signal transduction, and the DNA damage response[1, 2]. Ubiquitylation is catalyzed by the concerted action of ubiquitin (Ub)-activating (E1), Ub-conjugating (E2), and Ub-ligating (E3) enzymes. Deubiquitylating enzymes (DUB) antagonize ubiquitylation by removing Ub modifications from their substrates. Ub can be covalently conjugated to substrates in several ways: as a single Ub conjugated to a single (monoubiquitylation) or multiple sites (multiple monoubiquitylation), or as polymeric chains (polyubiquitylation). Different Ub chains are formed through isopeptide linkages using seven internal lysine (K) residues, as well as its N-terminal methionine (M1). Effector proteins harboring Ub-binding domains (UBDs) function as readers/decoders by discriminating specific Ub linkages[3]. In addition to homotypic chains, cells contain heterotypic Ub chains in which multiple linkages form mixed or branched chains. Furthermore, Ub undergoes phosphorylation and acetylation at specific S/T and K residues[4].

Accumulating evidence indicates that linkage type, length, and chemical modification work in concert to affect the topology and dynamics of Ub chains and direct substrates to distinct biological pathways[4–6]. In particular, linkage type is a critical determinant of chain function. For example, it is widely accepted that K48-linked chains function as targeting signals for proteasomal destruction, whereas K63-linked chains are generally involved in signal transduction, DNA repair, and trafficking of membrane proteins; other linkage types also have distinct cellular functions. In contrast, despite its fundamental importance, our knowledge regarding the functional relevance of Ub chain length remains limited. Earlier in vitro studies suggested that the proteasome recognizes K48-linked tetraubiquitin as the minimal targeting signal, and that binding strength increases markedly as chain length increases up to octaubiquitin[7]. However, more recent studies showed that monoubiquitylation and multiple short Ub chains also constitute efficient proteasomal targeting signals[8–10]. In endocytosis and endosomal targeting, the relative importance of monoubiquitylation and K63-linked polyubiquitylation of receptor proteins remains unclear[11].

To understand the biological significance of various Ub chain structures, it is essential to determine the linkage types, modifications, and lengths of endogenous, substrate-linked chains. Recent advances in mass spectrometry (MS) and antibody-engineering technologies allow us to determine and quantitate Ub linkages and PTMs in complex biological samples[12]. In contrast, the lengths of substrate-attached Ub chains have only been determined by analyzing their gel mobility. However, since most endogenous substrates have multiple ubiquitylation sites, and attached chains might have heterogeneous lengths[13, 14], more comprehensive and accurate techniques are required. Here we describe a novel biochemical method for determining Ub chain length, "Ub chain protection from trypsinization (Ub-ProT)." By combining this method with quantitative MS analysis, we identified the length and composition of Ub chains in yeast, and of ligand-activated epidermal growth factor receptor (EGFR) in mammalian cells.

## Results

### Establishing a method for determining Ub chains.
Because Ub can form polymeric chains, a given composition of ubiquitylation can form numerous structures, e.g., a substrate protein bearing four Ubs can form five distinct topologies even if linkage types and branching are not considered (Fig. 1a). Thus, the gel mobilities of ubiquitylated proteins do not accurately reflect individual chain Ub lengths. Analysis of Ub chains cleaved from substrate proteins at the proximal Ub moiety would be the optimal way to determine chain length. DUBs could be used for this purpose, but unfortunately, the known DUBs do not discriminate linkage positions. Although the proteasomal DUB Rpn11 can remove Ub chains by cleaving the proximal Ub, the reaction is coupled to substrate unfolding by ATPase subunits[15]. Therefore, we designed an alternative approach using trypsin and a Ub chain protector. A previous study showed that Ub is specifically cleaved at Arg74 by trypsin digestion under native conditions. This cleavage occurs for all different Ub linkage types[16] (Fig. 1b, i). However, if substrate-attached chains are masked by a Ub chain protector, intact polyubiquitin chains should remain after trypsinization, allowing the substrate-bound poly-Ub chains to be easily analyzed using a gel-based assay. We named this approach Ub-ProT (Fig. 1b, ii). As a Ub chain protector, we used the tandem Ub-binding entity (TUBE), a high-affinity probe for Ub chains[17]. TUBE is an artificial protein, consisting of four repeats of the Ub-associated (UBA) domain of human RAD23A or UBQLN1. The UBQLN1 TUBE binds K48- and K63-linked tetraubiquitins with a $K_d$ of 9 and 0.7 nM, respectively. To prevent trypsin digestion, we constructed trypsin-resistant (TR)-TUBE consisting of a biotin tag, a hexahistidine tag, and six tandem repeats of the UBQLN1 UBA domain in which the Arg residues were replaced by Ala[18, 19] (Supplementary Fig. 1). We confirmed that TR-TUBE could efficiently pull down all eight types of di-Ubs (Supplementary Fig. 2a). Then, we tested whether free K48-, K63-, M1-, and K11-linked Ub chains, K48/K63 branched chains, and di-Ubs consisting of all eight different linkage types could be protected from trypsin digestion by TR-TUBE (Fig. 1c, d; Supplementary Figs. 2–5). We optimized the amount of trypsin and reaction time for complete cleavage of K48-linked Ub chains (Supplementary Figs. 2b and 3). This condition was sufficient to digest all unprotected Ub chains to at least monomers. TR-TUBE protected all chain types from digestion (Fig. 1c, d; Supplementary Fig. 2c). Greater than 90% of input K11-, K48-, K63-, and M1-linked di-Ubs were protected from trypsinization, and 40–60% of K6-, K27-, K29-, and K33-linked chains were protected (Supplementary Figs. 2c and d; Supplementary Data 1). TR-TUBE could also protect K48/K63 branched chains from digestions, although the protection efficiency was lower than for homogeneous chains (Fig. 1d; Supplementary Fig. 5).

Since our TR-TUBE construct consists of six Ub-binding domains, one concern about our method was that it might only protect Ub chains up to hexamers. However, as seen in Fig. 1c, TR-TUBE could efficiently protect much longer chains, suggesting that multiple molecules of TR-TUBE can bind to a single chain to restrict trypsin accessibility. To further examine whether the number of Ub-binding domains in TR-TUBE may artificially influence the length of protected Ub chains, we constructed TR-TUBEs with four and eight Ub-binding domains, and compared the lengths of protected Ub chains (Supplementary Fig. 4). All three different TR-TUBE constructs protected M1-linked chains equally, and protected chains were indistinguishable from untrypsinized control chains.

We next applied the Ub-ProT method to tetraubiquitin-fused Sic1 (4×Ub-T7-Sic1) as a model substrate with a defined Ub chain length (Fig. 2a). The tetraubiquitin of 4×Ub-T7-Sic1 was digested to Ub monomers by trypsinization, but almost completely protected in the presence of TR-TUBE. In contrast, the same construct containing the I44A mutation in each Ub moiety (4×UbI44A-T7-Sic1) was completely digested, validating our method. Several Ub enzymes are efficiently self-ubiquitylated in vitro: Cdc34 is self-ubiquitylated with K48-linked chains[20], Rsp5 is self-ubiquitylated with K63-linked chains[21], and MBP-tagged Parkin is multiply monoubiquitylated[22]. In each case,

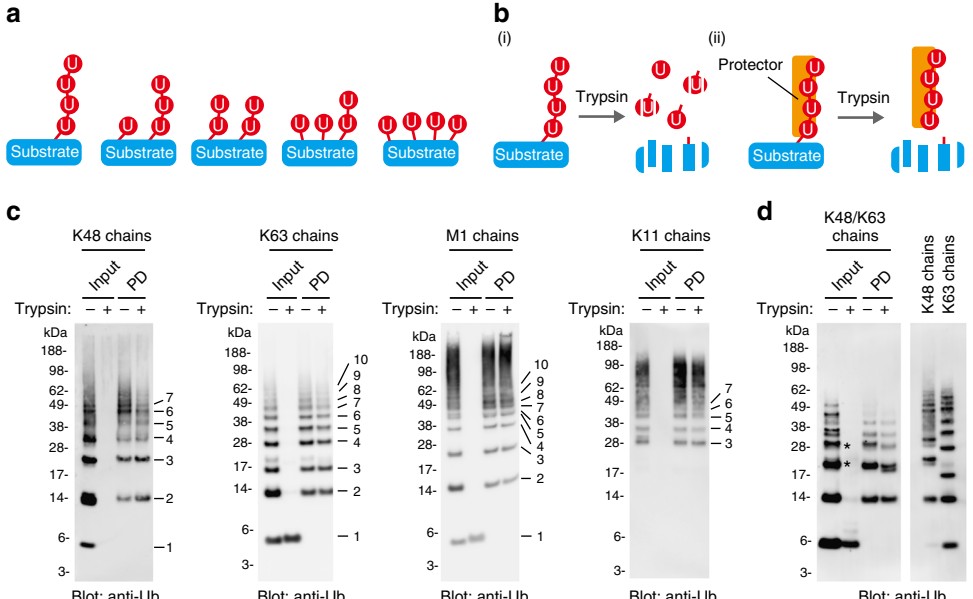

**Fig. 1** Establishment of a method for determining ubiquitin chain length. **a** Different combinations of ubiquitylation with four ubiquitin molecules. Ubiquitins are depicted as red circles. This schematic does not take into account different linkage types or branching chains, which greatly increase the number of combinations. **b** Conceptual schematic of the "ubiquitin chain protection from trypsinization (Ub-ProT)" method. (i) Trypsinization under non-denaturing condition results in near-complete digestion of substrate proteins and partial digestion of ubiquitin chains, (ii) whereas a protector such as TR-TUBE protects ubiquitin chains from digestion. **c** Ub-ProT assay of free ubiquitin chains. K48-linked (left), K63-linked (center left), M1-linked (center right), and K11-linked (right) chains were pulled down with TR-TUBE (PD) and subjected to trypsinization. Ubiquitin was probed with anti-ubiquitin antibody. Positions of ubiquitin chains with different lengths are numbered. Monomeric Ub is more resistant to trypsinization compared to Ub chains, and sometimes remains after trypsinization of input samples (lane 2 in the middle two gels). **d** Ub-ProT assay of K48/K63 branched chains. K48/K63 branched chains were captured by TR-TUBE, subjected to trypsinization, and detected using anti-ubiquitin antibody. Positions of ubiquitin chains with different lengths are numbered. Asterisks denote branched ubiquitin chains (Supplementary Fig. 5)

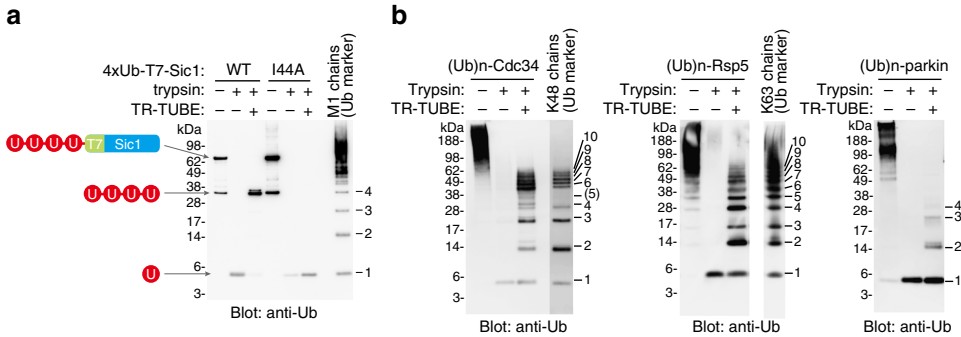

**Fig. 2** Ub-ProT protects substrate-attached ubiquitin chains in vitro. **a** Ub-ProT assay of tetraubiquitin-fused Sic1. M1-linked tetraubiquitin-fused T7-Sic1, cleaved tetraubiquitin, and ubiquitin monomer are indicated. A ubiquitin mutant (I44A in each ubiquitin moiety, resulting in a defect in TR-TUBE binding) was used as a control. Unanchored M1 chains were used as ubiquitin markers. **b** Ub-ProT assay of ubiquitylated proteins. Self-ubiquitylated GST-Cdc34 (Ubn-Cdc34, left), self-ubiquitylated GST-Rsp5 (Ubn-Rsp5, middle), and self-ubiquitylated MBP-Parkin (Ub-Parkin, right) were subjected to Ub-ProT. Free ubiquitin chains were used as markers

ubiquitylated proteins were detected as a smear at high molecular weight, which disappeared following trypsinization (Fig. 2b). In the presence of TR-TUBE and trypsin, the protein substrate was digested leaving typical Ub ladders. Comparison with free Ub chains (used as a length marker) revealed that Cdc34 and Rsp5 were modified with K48- and K63-linked chains, respectively, of up to ~10-mers (Fig. 2b, left and middle). In contrast, self-ubiquitylated Parkin was modified with monoubiquitin and, to a lesser extent, short Ub chains (Fig. 2b, right). Because TR-TUBE captured almost all ubiquitylated Parkin, we concluded that TR-TUBE can bind not only Ub chains but also multiply monoubiquitylated substrates.

**Measurement of chain length of yeast ubiquitylated proteins.** To characterize global Ub chain lengths in an entire organism, we next investigated the mean length of substrate-attached Ub chains in yeast lysates. In this experiment, we used a drug-sensitive *pdr5* mutant, which increases sensitivity to the proteasome inhibitor MG132. This allowed us to compare Ub chain lengths in control lysates and in lysates from yeast in which proteasomal activity was inhibited by MG132[23]. Soluble lysates were prepared from exponentially growing cells cultured with or without MG132, and ubiquitylated proteins were captured and pulled down by TR-TUBE. Immunoblotting with anti-Ub antibody revealed that TR-TUBE was unable to pull down Ub monomers, but otherwise

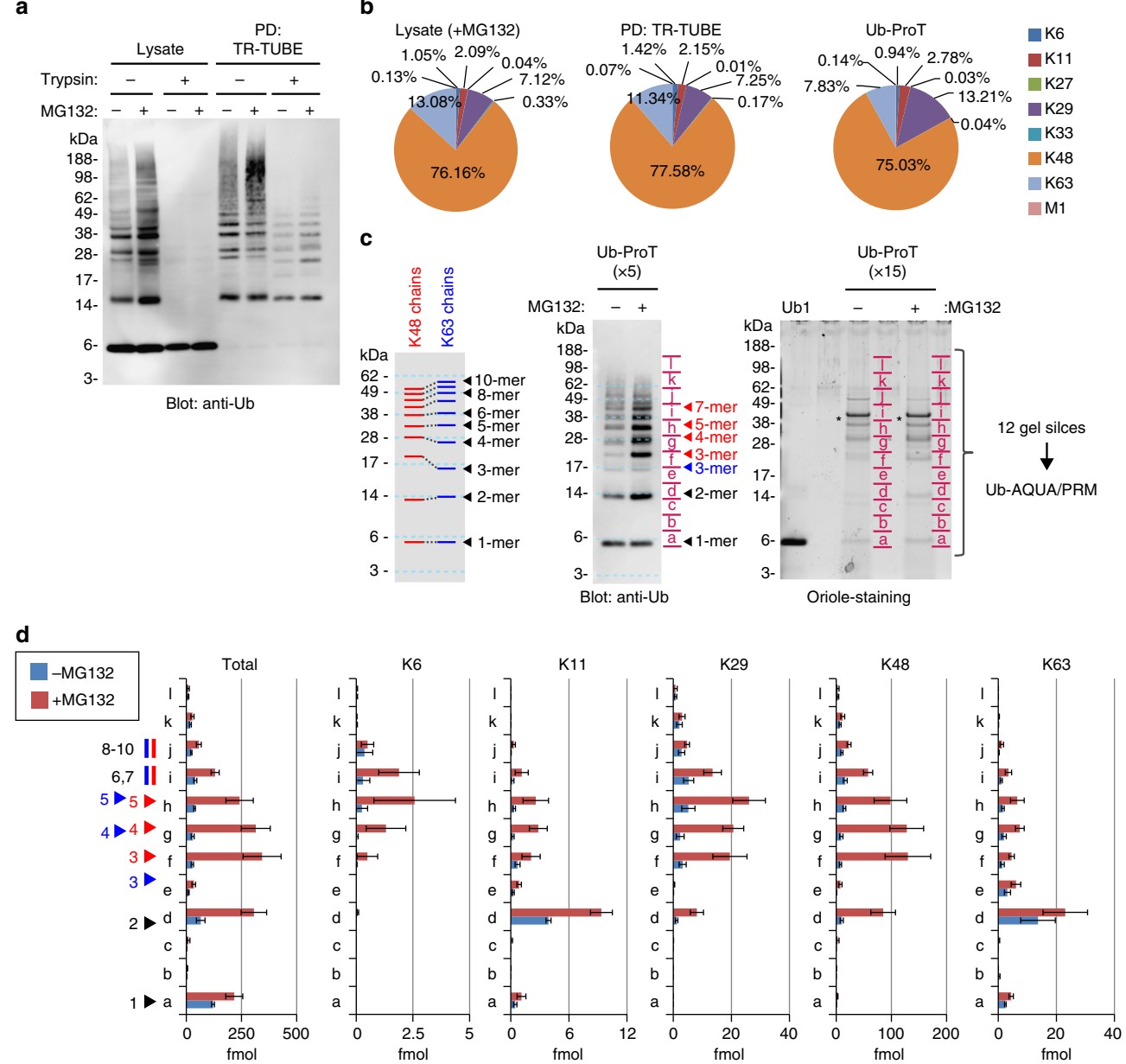

**Fig. 3** Length distribution of substrate-attached ubiquitin chains in yeast-soluble lysate. **a** Ub-ProT analysis of cellular ubiquitylated proteins. Exponentially growing cells were treated with or without 100 μM MG132 for 4 h. Ubiquitylated proteins in lysates were captured by TR-TUBE and subjected to trypsinization. **b** Composition of ubiquitin linkages in lysate, TR-TUBE-captured proteins, and Ub-ProT sample. For MG132-treated wild-type cells ($n = 5$) and sample pulled down (PD) with TR-TUBE ($n = 4$), the gel region above 49 kDa was subjected to Ub-AQUA/MS analysis (Supplementary Data 2). For Ub-ProT samples ($n = 3$), sum of Ub linkages quantified in **d** was represented (Supplementary Data 3). **c** Estimation of chain length of Ub-ProT samples. Gel mobility of free K48- and K63-linked chains (left; see Fig. S3 in detail). Anti-ubiquitin blot (middle) and protein staining (right) of the Ub-ProT sample. Here the amount of material analyzed was 5- or 15-fold higher than that in **a**, for the anti-Ub blot and protein staining, respectively. Gel regions subjected to MS quantitation are indicated by red letters (a–l). The position of the ubiquitin monomer (Ub₁) was defined as the gel fraction "a." Asterisks denote TR-TUBE. **d** Length distributions of total ubiquitin and five major linkages at steady state or following MG132 treatment. Gel fractions in **c** were analyzed by quantitative mass spectrometry (mean ± s.e.m.; $n = 3$ biological replicates; Supplementary Data 3). Relative positions of K48- and K63-linked chains are labeled in blue and red, respectively

captured endogenous ubiquitylated proteins efficiently (Fig. 3a, lanes 1 and 5). MG132 treatment increased the intensity of various ubiquitylated bands in both lysates and TR-TUBE pull-downs (lanes 2 and 6), indicating that ubiquitylated substrates accumulated upon proteasomal inhibition. We also compared the compositions of Ub linkages between lysate and pulled-down samples by quantitative MS analysis and Ub-absolute quantification (AQUA)/parallel reaction monitoring (PRM)[24]. In lysates,

K48 and K63 linkages were predominant, and the abundance of other linkages were as follows: K29 > K11 > K6 > M1 ≅ K27 ≅ K33 (Fig. 3b; Supplementary Data 2). Linkage compositions were similar between lysates and pulled-down samples, confirming that TR-TUBE does not generate significant linkage bias during pull-downs. Upon trypsinization of unprotected lysates, signals from Ub conjugates completely disappeared (Fig. 3a, lanes 3 and 4), while trypsinization of TR-TUBE-captured proteins generated a

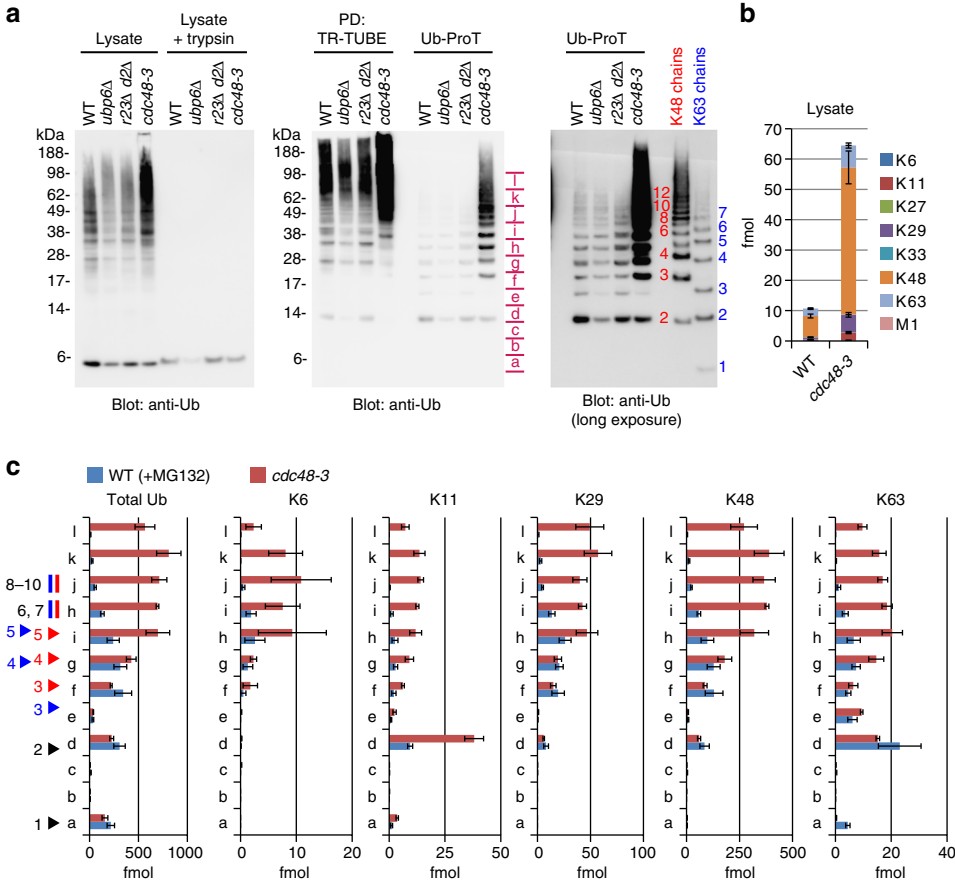

**Fig. 4** Cdc48/p97 regulates ubiquitin chain length. **a** Ub-ProT analysis of UPS-related mutant cells. Soluble lysate from wild-type (WT), *ubp6Δ*, *rad23Δ dsk2Δ* (*r23Δ d2Δ*), or *CDC48* mutant (*cdc48-3*) cells were subjected to Ub-ProT analysis. Gel regions subjected to MS quantitation are indicated by red letters (a–l). Free ubiquitin chains were used as markers. **b** Absolute amounts of ubiquitin linkages in lysate from MG132-treated and *cdc48-3* cells. The gel region above 49 kDa, where ubiquitylated proteins were detected, was subjected to Ub-AQUA/MS analysis (mean ± s.e.m.; *n* = 5 biological replicates; Supplementary Data 5). **c** Length distributions of total ubiquitin and five major linkages in MG132-treated wild-type or *cdc48-3* cells. Gel fractions in **a** were analyzed by quantitative mass spectrometry (mean ± s.e.m.; *n* = 3 biological replicates; Supplementary Data 6)

Ub chain ladder (Fig. 3a, lanes 7 and 8). In addition, different Ub chains with three or more Ubs exhibited different gel mobilities (Fig. 3c; Supplementary Fig. 6). Comparison with free Ub chains revealed that the lengths of the substrate-attached Ub chains were mainly in the monomer to heptamer range (Fig. 3c, left and middle). Although MG132 treatment increased the amounts of chains of each length, maximal lengths were unchanged.

To investigate the relationship between linkage types and chain lengths, we cut gel lanes into 12 pieces to separate monomers and different-length Ub chains (Fig. 3c, right). The resultant gel slices were subjected to Ub-AQUA/PRM (Fig. 3d; Supplementary Data 3). Because the abundances of the M1, K27, and K33 linkages were quite low (<50 attomoles: lower limit of Ub-AQUA/PRM; Supplementary Fig. 7; Supplementary Data 4), we focused on the five major linkage types detected in this experiment. In the absence of MG132, K48, the most abundant linkage type, was detected in di-Ub and longer chains. K11 and K63 linkages were mainly found in di-Ubs, whereas K6 and K29 linkages were mainly detected in longer chains. Proteasome inhibition by MG132 dramatically increased the abundance of each linkage type in each fraction. In particular, the abundance of the K29 and K48 linkages was elevated for dimer to heptameric chains, and the abundance of the K6, K11, and K63 linkages was elevated for longer chains (Fig. 3d). The K29 linkages in long chains may be involved in the Ub-fusion degradation (UFD) pathway, in which heterotypic chains with K29 and K48 linkages

are directed to proteasomal degradation[25]. It will be of great interest to determine whether the K6, K11, and K63 linkages mainly detected in the longer chains are homogeneous or heterogeneous. Moreover, it is noteworthy that nearly 50% of Ub was detected as monomers in untreated cells, and the amount of monomeric Ub increased by 1.8-fold in MG132-treated cells (Fig. 3d, total). TR-TUBE does not bind Ub monomers, suggesting that a significant portion of ubiquitylated cellular proteins are modified with multiple monoubiquitins[13] or with terminal monoubiquitin branches off of polyubiquitin chains. Together with a recent finding that ~20% of proteasome substrates are degraded upon monoubiquitylation[10], our results indicate that a wide range of Ub signals, including monoubiquitylation and chains of mainly dimer to heptamer length, function in proteasome targeting.

**Cdc48/p97 regulates Ub chain length.** We next sought to investigate factors that regulate the global chain length of ubiquitylated substrates. Ubp6, the proteasome-associated DUB, maintains cellular Ub levels by removing Ub chains from substrates en bloc, whereas Rad23 and Dsk2 shuttle substrates to the proteasome[26, 27]. The Ub-selective chaperone Cdc48 (p97/VCP in mammalian cells) is involved in endoplasmic reticulum (ER)-associated degradation (ERAD), autophagy, and other Ub-dependent processes[28]. In lysates, the levels of ubiquitylated

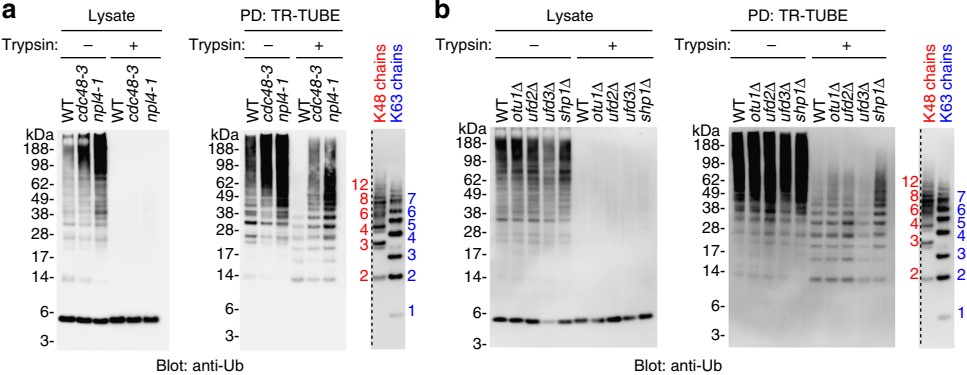

**Fig. 5** Ubiquitin chain elongation in *npl4-1* mutant. **a, b** Ub-ProT analysis of Cdc48 cofactor mutant cells. Soluble lysate from wild-type (WT), *npl4* mutant cells (*npl4-1*), *otu1Δ*, *ufd2Δ*, *ufd3Δ*, or *shp1Δ* cells were subjected to Ub-ProT analysis. Free ubiquitin chains were used as markers

substrates were reduced in *ubp6Δ* and *rad23Δ dsk2Δ* cells (Fig. 4a left; *ubp6Δ* and *r23Δ d2Δ*). This result is consistent with those of a previous study demonstrating that Rad23 and Dsk2 protect Ub chains from DUB[29]. In contrast, in *cdc48-3* mutant cells, we observed significant accumulation of ubiquitylated substrates, with elevated levels of five Ub linkage types (K6, K11, K29, K48, and K63) (Fig. 4a left and 4b; Supplementary Data 5). Strikingly, a subsequent Ub-ProT assay revealed that Ub chains were significantly elongated in the *cdc48-3* mutant, but unaffected in *ubp6Δ* and *rad23Δ dsk2Δ* cells (Fig. 4a, middle and right). Linkage quantification suggested that the elongated chains in *cdc48* cells contained all five major linkages (Fig. 4c; Supplementary Data 6). Because chain elongation was not observed in proteasome-inhibited cells, this effect was not simply due to the accumulation of ubiquitylated proteins. Instead, since Cdc48/p97 is involved in segregation/remodeling of protein complexes, and major ubiquitylating enzymes build Ub chains on their substrates through processive elongation[30, 31], Cdc48-dependent segregation/remodeling likely inhibits processive Ub chain elongation in substrate complexes. Substrate-attached Ub chains could be remodeled on Cdc48 complexes by multiple cofactors, such as Ub-binding cofactors (Ufd1, Npl4, Ufd3, and Shp1), the DUB Otu1, and E4 Ub elongation factor Ufd2[32, 33]. We investigated the role of these cofactors in regulation of substrate-attached Ub chain lengths (Fig. 5), and found that Ub chains were significantly elongated in a *npl4-1* mutant, which is defective for extraction of ubiquitylated membrane proteins[34] (Fig. 5a). Ub chains were also slightly elongated in *shp1Δ* cells, but unaffected in *otu1Δ*, *ufd2Δ*, and *ufd3Δ* cells (Fig. 5b). Previously, we reported that Ub-binding of Npl4 was completely abolished by the *npl4-1* mutation (G323S) in vitro[19]. Thus, we conclude that the recognition and segregation of ubiquitylated substrate by Cdc48 and Npl4 is important in regulating Ub chain lengths in cells.

**Chain topology in ubiquitylated EGFR.** Ubiquitylation of ligand-activated EGFR promotes clathrin-independent endocytosis and endosomal targeting of the receptor for lysosomal degradation[11]. EGFR is modified by multiple mono- and poly-ubiquitin chains linked through K11, K48, and K63 linkages[35], but the topology and functional relevance of these modifications are still under debate[36–38]. To explore this issue, we applied Ub-ProT to EGFR ubiquitylation. For this purpose, we generated HeLa cells stably expressing EGFR-EGFP-3×FLAG from the *AAVS1* locus (Fig. 6a). Upon EGF treatment (100 ng ml[−1]), EGFR-EGFP-3×FLAG was rapidly ubiquitylated, and was subsequently degraded after 60 min (Fig. 6b). Microscopic analysis confirmed that the tagged EGFR initially localized to the plasma membrane, but moved to endosomes after 15 or 30 min of EGF

stimulation, and then translocated to lysosomes after 1 h (Supplementary Fig. 8). Ub-AQUA/PRM analysis revealed that ubiquitylated EGFR at the 5-min time point contained the following Ub chain composition: K63 linkage (~50% of total Ub), K48 linkage (~4%), K11 linkage (~2%), and mono/endocap Ub (~40%) (Fig. 6c; Supplementary Data 7). These results are consistent with those from a previously published study[35]. We next subjected the ubiquitylated EGFR to Ub-ProT and observed a Ub ladder corresponding to K63-linked chains of four to six Ub molecules specifically in EGF-treated cells (Fig. 6d, lane 5). To determine whether these were true K63 chains, we used linkage-selective DUBs. Recently, Komander et al. developed a versatile method for analyzing the higher-order architecture of heterotypic chains by linkage-selective DUBs, termed "Ub chain restriction (UbiCRest)"[39]. Among the DUBs, we used K48-chain-selective OTUB1, K63-chain-selective AMSH, and non-selective USP2. As expected, the Ub ladder was completely digested by treatment with AMSH or USP2, but not with OTUB1 (Fig. 6d, lanes 6-8). Thus, ligand-activated EGFR is primarily modified with K63-linked tetra- to hexa-ubiquitins. When present, K48 and K11 linkages are formed at sites distal to the K63 chains.

To further investigate the relationship between EGFR trafficking and Ub chain length, we performed Ub-ProT at various times after EGF stimulation (Fig. 6e). Signals for monoubiquitylation, which may consist of multiple monoubiquitylations on the substrate, or of multiple monoubiquitylations that branch off of chains, and Ub chains were elevated after 5 min of EGF treatment, and reached a maximum after 30 min, suggesting that EGFR is continuously ubiquitylated from the plasma membrane to the endosome. After 60 min of stimulation, the intensities of Ub bands decreased, consistent with deubiquitylation at multivesicular bodies, and lysosomal degradation. Thus, although certain additional bands were also detected, the major Ub modifications to EGFR were multiple monoubiquitylations and K63-linked polyubiquitylation during the trafficking process (Fig. 6f). Consistent with this result, recent studies suggested that the K63-linked chains function in endosomal sorting rather than internalization of EGFR[36, 40]. It remains unclear why the K63 chains had intermediate length, i.e., why they were tetra- to hexa-ubiquitins rather than di- or tri-ubiquitins. One possible explanation is that long chains may be advantageous for collaborative recognition by the highly organized ESCRT complexes (0, I, II, and III), which contain multiple UBDs[41].

## Discussion

In this study, we developed a novel method for determining the chain length of ubiquitylated substrates. By combining this method (Ub-ProT) with MS-based Ub quantitation (Ub-AQUA/

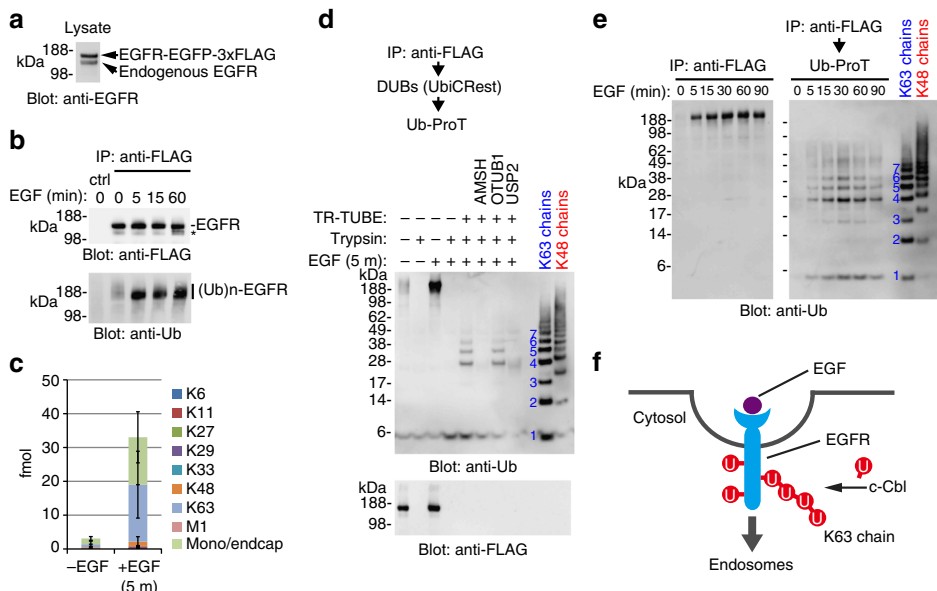

**Fig. 6** Ubiquitin chain length and composition of ligand-activated EGFR. **a** Anti-EGFR blot of HeLa cells stably expressing EGFR-EGFP-3×FLAG. **b** Ubiquitylation of EGFR. After 100 ng ml⁻¹ EGF treatment, EGFR-EGFP-3×FLAG was immunoprecipitated with anti-FLAG antibody and subjected to immunoblotting with indicated antibodies. Parental wild-type HeLa cells were also analyzed (ctrl). Degraded EGFR is indicated by an asterisk. **c** Absolute quantification of ubiquitin linkages in immunoprecipitated EGFR-EGFP-3×FLAG. The gel region above 120 kDa was subjected to Ub-AQUA/MS analysis (mean ± s.d.; $n = 2$ biological replicates; Supplementary Data 7). **d** Ub-ProT analysis of ubiquitylated EGFR combined with UbiCRest assay. Immunoprecipitated EGFR was treated with K63 linkage-specific AMSH, K48 linkage-specific OTUB1, or nonspecific USP2, followed by Ub-ProT. **e** Time-dependent changes of ubiquitin chains on EGFR. After EGF treatment, EGFR-EGFP-3×FLAG was immunoprecipitated at the indicated times and analyzed by Ub-ProT. **f** A possible model for ubiquitin chain architecture of ubiquitylated EGFR

PRM), we determined for the first time the global Ub chain lengths in endogenous ubiquitylated substrates from soluble yeast lysates (Fig. 3). At steady state, K48, K6, and K29 linkages were predominantly incorporated in dimeric to heptameric chains, whereas K11- and K63-linked chains existed mainly in dimeric chains. Surprisingly, maximum chain lengths were only slightly altered by proteasome inhibition, implying the existence of a proteasome-independent mechanism for regulating chain length. This mechanism may require specific recognition by UBD-containing proteins. For example, UBD-containing proteins such as Rad23/Dsk2 bind Ub chains with appropriate lengths (in the 2–7-mer range), thereby protecting them from DUBs. Alternatively, the chain-elongating ability of ubiquitylating enzymes might be intrinsically limited in vivo, as suggested by in vitro studies[30, 31]. In this context, because attachment of a single Ub costs one molecule of ATP, restricting chain length may benefit the cell by limiting total energy consumption. On the other hand, we found that inactivation of Cdc48 and Npl4 caused a significant accumulation of elongated Ub chains (Figs. 4 and 5), suggesting that chain length is regulated by sequestration by the Cdc48 complex rather than limitations of ubiquitylating enzymes. Cdc48 regulates ubiquitylation of over a thousand proteins[42], explaining its global effects on yeast Ub chain lengths. Ub-ProT analysis also revealed the Ub chain length and composition of ubiquitylated EGFR in HeLa cells (Fig. 6). Although we also detected K48 and K11 linkages, combined analysis with UbiCRest suggested that a K63-linked chain of four to six Ubs is the core unit of the attached chains. Further combinations of Ub-ProT, Ub-AQUA/PRM, UbiCRest, and middle-down MS might allow us to investigate the Ub chain architectures of more complex chains, including the positions of linkage branching and PTMs.

In all of our Ub-ProT assays, we used TR-TUBE as the chain protector. The advantages of TR-TUBE are its extremely tight binding to Ub chains, and its ability to bind in tandem to bind and protect long Ub chains. However, it is important to note the

limitations to our method. While we demonstrate that Ub-ProT using TR-TUBE can bind and protect all Ub linkage types, there is up to a twofold variability in the amount of protection. While 90% of K11-, K48-, K63-, and M1-linked dimeric Ub is protected by Ub-ProT using TR-TUBE, K6-, K27-, K29-, and K33-linked Ubs are protected at 40–60% efficiency (Supplementary Fig. 2). Therefore, our method may slightly underestimate the occurrence of certain linkage types. In addition, while TR-TUBE does not bind free monomeric Ub, we see some monomeric Ub when we analyze chain length and composition type after Ub-ProT treatment of lysates (Figs. 3d and 4c). We propose that most of this monomer is generated from multiply monoubiquitylated substrates, such as multiply monoubiquitylated Parkin, which is bound by TR-TUBE (Fig. 2b). However, there are trace amounts (<5%) of K11- and K63-linked Ub in the monomer fraction (Figs. 3d and 4c), indicate that at least some of the monomeric Ub is generated from a slight degradation of K11 and K63 chains that escaped protection by TR-TUBE. Furthermore, it is possible that some of the unlinked monomeric Ub observed may be generated from the trimmed terminal ends of chains instead of from multiple monoubiquitylations of substrates. Finally, while we demonstrate that Ub-ProT can also protect branched Ub chains (Fig. 1d), again this protection is somewhat reduced compared to homogeneous chains (Fig. 1c), raising the possibility that our method may underestimate the occurrence of branched chains.

Despite these caveats, our characterization of global yeast Ub chain lengths, regulation of chain lengths by Cdc48 and Npl4, and modification of EGFR upon EGF treatment indicate that Ub-ProT using TR-TUBE, or a novel chain protector, will be useful for the study of Ub chain lengths.

## Methods

**Plasmids**. *TUBE expression plasmids*: We modified a previously reported high-affinity probe for Ub chains, TUBE, which consists of four tandem repeats of the UBA domain of UBQLN1[17]. To capture ubiquitylated proteins more efficiently and

prevent trypsinization to allow MS analysis, we mutated all Arg residues in the UBA domain to Ala and fused four to eight repeats together[18]. Next, the gene encoding TR-TUBE was inserted into a modified pRSET-A vector (Life Technologies), which contains a hexahistidine tag and Cys residue for purification and biotinylation, respectively. The protein-coding sequence of TR-TUBE is shown in Supplementary Fig. 1b. Our TR-TUBE constructs will be added to Addgene.

4×Ub-Sic1 expression plasmids. To construct tetraubiquitin-fused Sic1, we first generated pET15b-4×Ub and pET15b-4×UbI44A, in which four Ubs were tandemly fused without a linker, followed by T7-Sic1$^{PY}$ inserted using the In-Fusion cloning kit (Takara). The template plasmid, pET15b-1×Ub, was kindly provided from Dr. K. Sakamoto (RIKEN institute, Japan). The resultant plasmids, pET15b-4×Ub-Sic1 and pET15b-4×UbI44A-Sic1, express 4×Ub-T7-Sic1$^{PY}$-His$_6$ and 4×UbI44A-T7-Sic1$^{PY}$-His$_6$, respectively.

DUB expression plasmids. To construct bacterial expression vectors for linkage-specific DUBs, cDNAs codon-optimized for bacterial expression of human AMSH and OTUB1 (Eurofins Genomics) were cloned into the pGEX6P1 expression vector (GE Healthcare). The bacterial expression vector for USP2, pET15b-USP2cc, was a gift from Dr. R. Baker[43].

TALE nuclease and EGFR plasmids. To generate HeLa cells stably expressing EGFR-EGFP-3×FLAG, we used the TALE nuclease (TALEN) method. AAVS1 site-specific TALEN plasmids were constructed as described previously[44] using TALE monomer template plasmids and TALEN backbone plasmids (Addgene #32185-32192). To construct donor vector pZDonor-AAVS1-EGFR-EGFP-3×FLAG, the sequences encoding the EF1 promoter, human EGFR cDNA (NM_005228), EGFP, 3×FLAG, and SV40 polyA were tandemly inserted into pZDonor-AAVS1-Neo, in which the puromycin resistance gene of pZDonor-AAVS1-puromycin (Sigma-Aldrich) was replaced with the G418 resistant gene.

**Yeast strains and media.** Saccharomyces cerevisiae strains used in this study are listed in Supplementary Table 1. All strains are isogenic to W303 or BY4741. Standard media and genetic techniques were used to manipulate yeast strains[45]. A deletion mutant of PDR5 was used to increase sensitivity to the proteasome inhibitor MG132[23]. Yeast cells were grown at 28 °C in SC medium (0.67% yeast nitrogen base without amino acids, 0.5% casamino acids, 2% glucose, 10 mM potassium phosphate (pH 7.5), 400 mg l$^{-1}$ adenine sulfate, 10 mg l$^{-1}$ uracil, and 20 mg l$^{-1}$ tryptophan).

**Cell lines and culture.** HeLa cells (laboratory stock of M. Komada)[46] were grown in Dulbecco's modified Eagle's medium (DMEM; Sigma-Aldrich) supplemented with 10% fetal bovine serum (FBS) and penicillin/streptomycin in a humidified 37 °C incubator with 5% CO$_2$. To introduce EGFR-EGFP-3×FLAG to the AAVS1 locus in HeLa cells, pTALEN-AAVS1 (left), pTALEN-AAVS1 (right), and pZDoner-EGFR-EGFP-3×FLAG expression plasmids were co-transfected into HeLa cells using a NEPA21 Super Electroporator (NEPAGENE). Cells were maintained in DMEM supplemented with 10% FBS and penicillin/streptomycin. After 2 days, culture medium was replaced with DMEM supplemented with 10% FBS, 0.5 mg ml$^{-1}$ G418, and penicillin/streptomycin. Cells were maintained for 3–5 days, and clones that expressed the protein were selected under a confocal microscope and maintained for another 1–2 weeks. Media containing G418 were replaced every 2 days. For stimulation with EGF, cells were serum-starved in DMEM supplemented with 0.5% FBS for 24 h, followed by 2 h in 0% FBS/DMEM. Cells were subsequently incubated with human EGF (100 ng ml$^{-1}$; PeproTech, Rocky Hill, NJ) or Texas Red-conjugated EGF (100 ng ml$^{-1}$; Thermo Scientific) at 37 °C.

**Expression and purification of TR-TUBE.** TR-TUBE was expressed in Escherichia coli Rosetta2 (DE3) for 15 h at 22 °C. Cells were lysed by passage through a pre-cooled French pressure cell (Ohtake Works) in lysis buffer (50 mM sodium phosphate (pH 7.0), 300 mM NaCl, 10% glycerol, and 1 mM Tris [2-carboxyethyl] phosphinehydrochloride), and the lysate was clarified by 30-min centrifugation at 29 300 × g. The supernatant was incubated with TALON resin (Clontech), and TR-TUBE was eluted with elution buffer (50 mM sodium-HEPES (pH 7.1), 100 mM NaCl, and 200 mM imidazole). Then, TR-TUBE was biotinylated with EZ-link Maleimide-PEG2-Biotin (Thermo Scientific), and further purified by gel filtration on Superdex 75 10/100 GL (GE Healthcare), pre-equilibrated with 50 mM HEPES (pH 7.5), 100 mM NaCl, and 10% glycerol. Purified biotin-TR-TUBE was divided into small aliquots and stored at −80 °C. Biotinylated 6×TR-TUBE was used throughout the study unless otherwise noted.

**Preparation of Ub chains and ubiquitylated proteins.** For Fig. 1c and Supplementary Fig. 2, the following commercial Ub reagents were obtained from Boston Biochem; K48-linked polyubiquitin chains (1–7), K63-linked polyubiquitin chains (1–7), and di-ubiquitins. K11-linked polyubiquitin chains were obtained from UBPBio. For other figures, we prepared unanchored M1-, K48-, and K63-linked poly-ubiquitin chains using petite-LUBAC, UBE2K/E2-25K, and UBE2N/UBC13-UBE2V1, as described previously[47–49]. For K48/K63 branched chains, 2 μg of bovine Ub (Sigma-Aldrich), 50 ng of human His$_6$-E1 (Boston Biochem), and 400 ng of Ubc13-Uev1a were incubated in 20 μl of reaction buffer (50 mM Tris-HCl (pH 7.5), 5 mM MgCl$_2$, 2 mM ATP, and 0.5 mM dithiothreitol (DTT)) at 37 °C for 2 h. Then, 400 ng of E2-25K was added and incubated at 37 °C for 3 h. Self-

ubiquitylated GST-Cdc34 was prepared by incubating 100 μg ml$^{-1}$ GST-Cdc34 on glutathione-Sepharose 4B beads (GE Healthcare) for 15 h at 37 °C in the presence of 33 μg ml$^{-1}$ human His$_6$-E1, 500 μg ml$^{-1}$ bovine Ub in 20 mM Tris-HCl (pH 7.5), 10 mM MgCl$_2$, 0.1 mM DTT, and 2 mM ATP, as described previously[20]. Self-ubiquitylation of GST-Rsp5 was carried out by incubating 50 μg ml$^{-1}$ GST-WW-HECT[50] on glutathione-Sepharose 4B beads for 15 h at 28 °C in the presence of 6.25 μg ml$^{-1}$ human His$_6$-E1, 50 μg ml$^{-1}$ Ubc4, 500 μg ml$^{-1}$ Ub in 50 mM sodium-HEPES (pH 7.5), 100 mM NaCl, 10% glycerol, 10 mM MgCl$_2$, 1 mM DTT, and 5 mM ATP. Self-ubiquitylated MBP-Parkin was prepared by incubating 20 μg ml$^{-1}$ MBP-Parkin on Amylose resin (New England BioLabs) for 3 h at 32 °C in the presence of 1.6 μg ml$^{-1}$ human His$_6$-E1, 100 μg ml$^{-1}$ Ubc4, 50 μg ml$^{-1}$ Ub in 50 mM Tris-HCl (pH 8.8), 2 mM MgCl$_2$, 2 mM DTT, and 4 mM ATP, as described previously[22]. After the reactions, the beads were washed with PBS plus 0.05% Tween 20 (PBS-T) and stored at 4 °C.

**Expression and purification of DUBs.** Recombinant AMSH, OTUB1, and USP2 were prepared as described previously[39, 43]. In brief, His$_6$-USP2 was expressed in E. coli BL21 (DE3) harboring pET15c-USP2cc and purified on TALON resin. GST-AMSH and GST-OTUB1 were expressed in E. coli BL21 (DE3) harboring pGEX6P1-AMSH or pGEX6P1-OTUB1 and purified on glutathione-Sepharose 4B beads. The glutathione S-transferase (GST) tag was removed by PreScission Protease (GE Healthcare). DUB activity was verified by cleaving free K48- or K63-linked chains.

**Ub-ProT assay for in vitro substrates.** Because the trypsin sensitivities of proteins vary with their structural properties, the amount of trypsin was titrated in each experimental setup. Di-Ub chains (500 ng) and TR-TUBE (5 μg) were incubated overnight at 37 °C in 20 μl of trypsin solution (50 mM ammonium bicarbonate (AMBC), 0.01% Rapigest SF (Waters), and 5 ng μl$^{-1}$ trypsin (Trypsin Gold; Promega)). For unanchored polyubiquitin chains, M1-, K11-, K48-, K63-linked, or K48/K63 branched Ub chains (100 ng) were incubated with TR-TUBE (5 μg) for 1 h at 4 °C, followed by incubation for 45 min at 4 °C with Dynabeads MyOne Streptavidin C1 (1 mg, Life Technologies). Bead-bound TR-TUBE–Ub chains were pulled down with a magnet, washed three times with PBS-T, and then incubated overnight at 37 °C in 20 μl of trypsin solution (50 mM AMBC, 0.01% Rapigest SF, and 2.5 ng μl$^{-1}$ trypsin). Ub4-Sic1 protein (500 ng) and TR-TUBE (5 μg) were incubated overnight at 37 °C in 20 μl of trypsin solution (50 mM AMBC, 0.01% Rapigest SF, and 5 ng μl$^{-1}$ trypsin). For self-ubiquitylated E2 or E3s, Ub conjugates (1 μg) immobilized on beads were incubated at 37 °C with TR-TUBE (5 μg) in 20 μl trypsin solution (50 mM AMBC, 0.01% Rapigest SF, and 15–25 ng μl$^{-1}$ trypsin). The reaction was quenched by addition of NuPAGE LDS sample buffer.

**Ub-ProT assay for yeast lysates.** For Ub-ProT assay of yeast extracts, 30 OD$_{600}$ units of log-phase cells were harvested and lysed with glass beads in 300 μl of lysis buffer (50 mM Tris-HCl (pH 7.5), 100 mM NaCl, 10% glycerol, 10 μM MG132, 10 mM iodoacetamide, and 1× complete protease inhibitor cocktail (EDTA-free, Roche)). After centrifugation, the supernatant (100 μg) was incubated with TR-TUBE (10 μg) for 1 h at 4 °C, followed by incubation for 45 min at 4 °C with Dynabeads MyOne Streptavidin C1 (1 mg). Bead-bound TR-TUBE–polyubiquitylated proteins were pulled down with a magnet, washed three times with PBS-T, and then incubated overnight at 37 °C in 100 μl of trypsin solution (50 mM AMBC, 0.01% Rapigest SF, and 15 ng μl$^{-1}$ trypsin). We found that streptavidin was not digested by trypsin under this condition; therefore, the polyubiquitin chains were still retained on the beads via the TR-TUBE/streptavidin complex after trypsinization. After the beads were washed with PBS-T, the poly-ubiquitin chains were selectively eluted by 30-min incubation with 1× NuPAGE LDS sample buffer. The samples were directly subjected to electrophoresis on NuPAGE gels without boiling in order to avoid aggregation of polyubiquitin chains[39].

**Immunoprecipitation and Ub-ProT assay for EGFR.** After EGF stimulation, HeLa cells stably expressing EGFR-EGFP-3×FLAG were lysed in denaturing buffer (50 mM Tris-HCl (pH 8.0), 150 mM NaCl, 10% glycerol, 1% SDS, 0.5% deoxycholate, 40 μM bortezomib (LC Laboratories), 50 μM PR619 (Abcam), 10 mM iodoacetamide, 1× PhosSTOP (Roche), and 1× complete protease inhibitor cocktail) and sonicated on ice for 30-s intervals with 1-s short burst and 1-s rest at moderate power (UR21P, TOMY SEIKO). After centrifugation twice at 12 000 × g for 10 min, the supernatants were diluted 10-fold in 1% Triton X-100-containing lysis buffer (TX-100 buffer: 20 mM Tris-HCl (pH 7.4); 100 mM NaCl; 1% Triton X-100; 1 mM EDTA; 10 mM iodoacetic acid; 40 μM bortezomib; 50 μM PR619; 1× PhosSTOP phosphatase inhibitor cocktails; and 1× complete protease inhibitor cocktail). Then, EGFR-EGFP-3×FLAG was immunoprecipitated from 500 μg of lysates with 25 μl of anti-FLAG M2-affinity gel (Sigma-Aldrich). To reduce the level of contaminating proteins, the beads were extensively washed: twice with TX-100 buffer; twice with TX-100 buffer containing 500 mM NaCl; and once more with TX-100 buffer. For Ub-AQUA/PRM analysis, the beads were incubated with 1× NuPAGE LDS sample buffer. For DUB assays, the EGFR-immobilized beads were washed once with DUB buffer (50 mM Tris-HCl (pH 7.5), 50 mM NaCl, and 5 mM DTT), and then incubated for 2 h at 37 °C with gentle rotation in the

presence of 0.7 μg of AMSH, 2 μg of OTUB1, or 2 μg of USP2 in 200 μl of DUB buffer. For Ub-ProT assays, the beads were washed three times with TX-100 buffer and incubated for 1 h at 4 °C with TR-TUBE (1 μg) in 100 μl of TX-100 buffer. Then, the beads were washed twice with TX-100 buffer, twice with detergent-free buffer (20 mM Tris-HCl (pH 7.4), 100 mM NaCl, and 1 mM EDTA), and twice with 50 mM AMBC. Finally, the beads were incubated overnight at 37 °C with gentle rotation in 100 μl of trypsin solution (50 mM AMBC, 0.01% Rapigest SF, and 5 ng μl$^{-1}$ trypsin). After centrifugation, the supernatants were mixed with 3× NuPAGE LDS sample buffer and subjected to SDS-polyacrylamide gel electrophoresis (SDS-PAGE) without boiling.

**SDS-PAGE and immunoblotting.** Proteins were separated by SDS-PAGE on 4–12% NuPAGE Bis-Tris gels (Life Technologies) and visualized with Oriole fluorescent gel stain (Bio-Rad) or Bio-Safe Coomassie Stain (Bio-Rad). For immunoblotting, the proteins were transferred to polyvinylidene fluoride membrane (GE Healthcare) on an XCell II Blot Module (Life Technologies). After verifying transfer by Ponceau S staining, we performed immunoblotting with the following antibodies: mouse monoclonal antibody against Ub (P4D1, horseradish peroxidase (HRP)-conjugated; used at 1:500 for immunoblotting; Santa Cruz Biotechnology); mouse monoclonal antibody against the FLAG-tag (M2, HRP-conjugated; 1:1000; Sigma-Aldrich); mouse monoclonal antibody against EGFR (6F1; 1:1000; MBL); rabbit monoclonal antibody against K48-linked (EP8589; 1:1000; Abcam); and rabbit monoclonal antibody against K63-linked (EPR8590-448; 1:1000; Abcam). HRP-conjugated goat anti-mouse Ig (1:10 000), used as a secondary antibody, was purchased from Jackson ImmunoResearch Laboratories. Immunoblots were developed using ECL Prim Western Blotting Detection Reagent (GE Healthcare), and analyzed on an ImageQuant LAS4000 (GE Healthcare). Uncropped blot images are shown in Supplementary Fig. 9.

**Ubiquitin-AQUA/PRM.** MS/MS-based absolute quantitation (AQUA) of Ub peptides by PRM was performed as described previously[24]. For the yeast lysates and TR-TUBE-pull-down samples shown in Figs. 3b and 4b, proteins (10 μg) were fractionated on NuPAGE gels with a short run (3 cm). The gel region corresponding to molecular weights >49 kDa was excised, diced into 1-mm$^3$ pieces, and subjected to trypsinization. For Ub-ProT samples, proteins were fractionated by NuPAGE gels with a full run (8 cm); gel lanes were cut into 12 fractions, starting at the position corresponding to the Ub monomer, using a grid cutter (2 mm long × 7 mm wide, Gel Company); the resultant slices were subjected to trypsinization. Digests were extracted by addition of 50 μl of 50% ACN/0.1% trifluoroacetic acid (TFA) and shaking for 1 h. The peptides were recovered into fresh Eppendorf tubes, and an additional extraction step was performed with 70% ACN/0.1% TFA for 30 min. The extracted peptides were concentrated using a speed-vac, spiked with Ub-AQUA peptides (K6, K11, K27, K29, K33, K48, K63, and M1 linkages, as well as ESTLHVLR (EST)), and then oxidized with 0.05% H$_2$O$_2$/0.1% TFA for 12 h at 4 °C as described[51]. For Ub-ProT samples, E. coli matrix (MassPREP, Waters) was added to the peptides samples (100 ng on column) to avoid nonspecific peptide adsorption. The peptides were analyzed in targeted MS/MS mode on a Q Exactive mass spectrometer coupled with an EASY-nLC 1000 liquid chromatograph and nanoelectrospray ion source (Thermo Scientific). The mobile phases were 0.1% formic acid (FA) in water (solvent A) and 0.1% FA in 100% ACN (solvent B). Peptides were directly loaded onto a C18 analytical column (ReproSil-Pur 3 μm, 75 μm inner diameter and 12 cm length, Nikkyo Technos) and separated using a 90-min three-step gradient (0–10% solvent B for 5 min, 10–30% for 70 min, and 30–80% for 5 min) at a constant flow rate of 300 nl min$^{-1}$. For ionization, 1.8 kV liquid junction voltage and 250 °C capillary temperature were used. The Q Exactive was operated by the Xcalibur software in target MS/MS mode, with an orbitrap resolution of 70 000 at $m/z$ 200, target automatic gain control values of $1 \times 10^6$, maximum ion fill times of 200 ms, isolation window of 2.0 $m/z$, and fragmentation by Higher-energy C-trap dissociation (HCD) with normalized collision energies of 28. Raw data were processed using the PinPoint software, version 1.3 (Thermo Scientific). Total Ub was determined from the sum of EST and K63-linked peptides. The raw data obtained and the inclusion list for PRM analysis, which contained target masses, charge states, and time windows for endogenous (light) and AQUA (heavy) peptides, are shown in Supplementary Data 2–7. We previously established standard curves to cover the range from 100 amol to 500 fmol for each Ub peptide[24]. We further determined the lower limit of quantitation for each Ub peptide (Supplementary Fig. 7a). We also confirmed that iodoacetamide used in preparation of yeast lysates did not affect Ub quantitation (Supplementary Fig. 7b).

**Fluorescence microscopy.** To monitor EGFR traffic, HeLa cells stably expressing EGFR-EGFP-3×FLAG were serum-starved in DMEM supplemented with 0.5% FBS for 24 h, followed by 2 h in 0% FBS/DMEM. Subsequently, cells were stimulated with 100 ng ml$^{-1}$ of Texas Red-conjugated EGF (Thermo Scientific) at 37 °C for the indicated times. Cells were fixed in 3% formaldehyde/PBS for 15 min on ice, permeabilized with 0.5% Triton X-100/PBS for 10 min, and blocked with 10% FBS/PBS. Cells were subsequently incubated with anti-EEA1 antibody (1 μg ml$^{-1}$; #E41120, BD Transduction Laboratories) in blocking buffer for 1 h at room temperature, washed three times with PBS, and incubated with Alexa Fluor 647-conjugated anti-mouse IgG antibody (1:1000; Invitrogen) in blocking buffer for 1 h

at room temperature. Nuclei were stained with 4′,6-diamidino-2-phenylindole. Fluorescence images were captured on a LSM710 laser-scanning confocal microscope (Carl Zeiss) equipped with a PLAN/APO ×63 oil objective and the ZEISS ZEN 2011 software.

**Data availability.** The authors declare that the RAW files of Ub-AQUA/PRM data have been deposited in the PeptideAtlas repository under data set ID PASS01132. All other data supporting the findings of this study are available within the manuscript and its Supplementary Files or are available from the corresponding author upon request.

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

## Acknowledgements

We thank Drs. Feng Zhang, Kazuhiro Iwai, Rohan T. Baker, Kensaku Sakamoto, Yusuke Sato, Noriyuki Matsuda, and Yoko Kimura for providing reagents. We also thank Dr. Hidehito Yoshihara for instruction in generating cell lines, and Dr. Junjiro Horiuchi for comments on the manuscript. This work was supported by MEXT/JSPS KAKENHI (JP15H06882 to H.T., JP24112003 to M.K., JP24112004 to F.O., JP21000012 to K.T., and JP24112008 and JP17H05681 to Y.S.), the Sumitomo Foundation for Basic Science Research (130479 to Y.S.), and the Takeda Science Foundation (to Y.S. and K.T.).

## Author contributions

Y.S. designed the study; H.T., D.B., N.A., A.K., and Y.S. performed experiments; H.T., D.B., M.K., F.O., K.T., and Y.S. wrote the manuscript. All authors reviewed the results and approved the final version of the manuscript.

## Additional information

**Competing interests:** The authors declare no competing financial interests.

