## [Peer Review File · Nature Communications]

Reviewers' comments:

Reviewer #1 (Remarks to the Author):

This manuscript describes a method that enables the identification of polyubiquitylation motifs in the yeast proteome using affinity purification and middle down proteomics. They used a previously reported trypsin-resistant tandem ubiquitin-binding entity(ies) (TR-TUBE) to bind modified substrates and prevent the cleavage of the polyubiquitin chains by trypsin (Yoshida et al., Proc Natl Acad Sci U S A. 2015, 112(15):4630-5). The resulting sample is loaded on SDS-PAGE and resolved based on polymeric ubiquitin chain linkages. The SDS-PAGE bands are subjected to in gel digested and the ubiquitylation linkages are quantified using synthetic peptides (AQUA). Using this method, the authors show the importance of Cdc48 in regulating polyubiquitin chain length. The authors also used their method to determine ubiquitin chain linkages found on EGFR upon ligand binding. The method described here is of potential interest to Nature Commun readership as it facilitates the identification of some polyubiquitin motifs. To fully appreciate the originality of this contribution, the authors must also contrast its benefits since this affinity purification approach was previous reported by the same group (Yoshida et al.). Minor comments are highlighted below.

1) On page 8, the authors mention “A previous proteomics study showed that yeast tryptic peptides are, on average, 8.4 amino acids in length. Thus, if the ubiquitylation sites were structurally hindered (i.e., if trypsin were unable to attack the proximal ubiquitins), the substrate-attached ubiquitin chains should converge to the individual chain sizes.” This phrase should be restructured or removed, as it leads to confusion.

Moreover, the ubiquitin found on the target protein is digested readily by trypsin since there is a commercial antibody that recognized the diglycine motif on ubiquitin modified peptides.

2) Figure 1d, Can the authors comment on the occurrence of intact ubiquitin after tryptic digestion?

3) Figure 2a. Why there is no ubiquitin monomer in the PD: TR-TUBE lanes but there is free ubiquitin in panel C (middle gel)?

Reviewer #2 (Remarks to the Author):

Tsuchiya et al. report on a novel method termed “ubiquitin chain protection from trypsination” (Ub-ProT) to determine the length of ubiquitin (Ub) chains on client proteins. In Ub-ProT ubiquitylated material is isolated by employing a protein termed TR-TUBE, which encompasses six repeats of a trypsin insensitive variant of the UBQLN1 UBA Ub binding domain. First, the authors demonstrate that TR-TUBE binds in vitro synthesised Ub chains linked via different lysine residues. The bound material is then treated with trypsin. Ub chains are protected from digestion by their interaction with the TR-TUBE whereas other proteins are removed. The authors then determine the size of the Ub chains by gel electrophoresis and analyse their composition by quantitative mass spectrometry. As a proof of concept the authors present data on the composition of Ub chains in yeast cell lysates and on the EGF receptor in EGF-treated HeLa cells.

General Points:

The presented study introduces a simple and potentially powerful method to analyse Ub modifications from total cell lysates. The experiments are of a more than adequate quality and in most cases contain appropriate controls. However, I am not convinced on the general versatility of the assay. An essential prerequisite for Ub-ProT is the robust protection of Ub chains by TR-TUBE from trypsin digestion. However, by comparing the amounts of di-Ub and of a lower molecular weight species that, to my opinion, represents mono-Ub in the TR-TUBE lanes in Figure S2c I get the impression that TR-TUBE shields di-Ub linked via distinct lysines from trypsination to a different degree. For example, K48-linked di-Ub appears to be less sensitive to the protease than K6- or K27-linked Ub (compare the amounts of di-Ub in the input and TR-TUBE lanes). This may be attributed to different binding affinities of these molecules or by diverging accessibility of cleavage sites in the bound state. In addition, the authors do not convincingly show that the binding to TR-TUBE fully prevents the access of trypsin to long poly-Ub chains of all linkage types (see also my comments below). Another critical issue of the assay is the amount of added protease. In their experiments, the authors determined the ideal trypsin concentration using only K48-linked di-Ub (Fig. S2b). In consequence, the experimental conditions allegedly favour the preservation of certain poly-Ub forms more than others, which complicates the analysis of complex samples. As the authors recognize, further problems arise from Ub chains containing heterogeneous linkage types or branching points. In summary, the assay can be individually optimized for the study of a defined poly-Ub modification but is in its current state probably less suitable to quantitatively analyse different poly-Ub species in parallel. Given these concerns and the issues listed below I do not recommend publication of this work in its present form (major revision).

Specific points:

Figure 1C: Why is mono-Ub completely digested by trypsin in the K48 sample, while it remains largely untouched in the K63 and M1 experiments? I also don't agree with the author's argumentation here. In the K48 experiment there is a considerable re-arrangement of poly-Ub in the PD lanes. Upon trypsin treatment the amount of high-molecular weight poly-Ub substantially decreases and at the same time there is an increase in the levels of di-Ub. The authors should employ longer K48- and K63-linked poly-Ub chains to ensure quantitative protection of longer poly-Ub chains by TR-TUBE.

Figure 3: Cdc48 mutants accumulate roughly six fold higher levels of poly-ubiquitylated material than the other yeast strains as assessed by Ub-AQUA/MS analysis (Fig. 3b). Thus, the increased length of Ub chains in the cdc48-3 samples (Fig. 3d) may be explained by insufficient protease capacity, given that a portion of poly-Ub is still sensitive to trypsin even in presence of TR-TUBE. Does the analysis of Ub chain length yield varying results, when trypsin concentration is changed (e.g. normalized to amount of Ub chains)?

The authors sometimes tend to over-interpret their results without considering alternative explanations. For example, the authors state that "Cdc34 and Rsp5 modified themselves with K48- and K63-linked chains, respectively up to 10-mer length" (page 7 lines 134-136). This statement is not fully correct. The authors should consider that longer chains may have not been completely covered by TR-TUBEs and thus have been partially degraded by trypsin. A closer view on Fig. 1c suggests that high molecular weight Ub chains may indeed not be fully protected from trypsin by TR-TUBE. Thus, a more careful phrasing is required. The authors also show that TR-TUBE does not bind mono-Ub but still they find a considerable amount of this species in their samples after trypsin treatment. According to their idea, this mono-Ub is derived from proteins containing multiple mono-Ub modifications. However, mono-Ub may also be generated by the trypsination of Ub chains that were not fully protected by TR-TUBEs. In fact, there are small amounts of peptides indicative for K11- and K63- (and traces of K48-) linked Ub found in fractions that correspond to mono-Ub (Fig. 2e & 3d). The authors agree that the binding of TR-TUBE to longer or branched Ub chains has not been systematically tested, and hence some mono-Ub may originate from partial decomposition of such structures. The authors also find that inactivation of Cdc48 increases the average length of some poly-Ub species. Their interpretation is that the segregation of substrate-ligase complexes by this AAA-ATPase terminates the ubiquitylation reaction and thereby restricts chain length. However, Cdc48 is known to associate with various Ub elongating and de-ubiquitylating enzymes, which were shown to shape the Ub landscape on substrates. Defects in Cdc48 function may affect the activity of these Ub modifying enzymes and thereby cause changes in the overall composition of poly-Ub chains.

Minor issues:

There are two types of asterisks in Fig. S1d that should be explained in the legend.

Reviewer #3 (Remarks to the Author):

Review of Tsuchiya et al. – A method for determining ubiquitin chain length and global architecture of protein ubiquitylation in cells. Nature Communications 2017

GENERAL COMMENTS:

In this manuscript, the authors present a new and quite ingenious method for determining Ub chain length on substrates *in vitro* and *in vivo*. This method uses Ub chain protection using a Trypsin-resistant TUBE and is termed Ub-ProT. After pulldown, tryptic digest, the power of mass-spectrometry (AQUA/PRM analyses) and careful gel-based analysis is used to reveal Ub chain length samples.

The method and idea are excellent and a big advance for the Ub field as it provides the field of ubiquitin signaling research with a new and potentially useful and valuable tool to understand the size and chain type of Ub modifications on cellular substrates, a task that to this day has remained difficult. The method will lead to a better understanding of endogenous, ubiquitin-modified substrates.

The manuscript is well written and the data is clear. With addressing the sole experimental concern for the study and some careful rephrasing to avoid misunderstandings and not oversell the work, I would support publication.

Below is a list of suggested improvements of the manuscript.

MAIN COMMENT requiring new experiments:

The TR-TUBE used throughout this study consists of 6 UBA domains. Although the authors observe longer chains, especially in the *cdc48-3* mutant strain, they observe a peak in chain length on substrates of ~6 Ub moieties in both unperturbed cells in figure 2 and in EGF-stimulated cells in figure 4. This reviewer is wondering if these observed chain lengths are true or whether they are a consequence of the capacity of the TR-TUBE to bind 6 Ub moieties, i.e., are the authors systematically underestimating the true length of the chains and the proportion of longer chains due to the fact that the TR-TUBE might not be proficient in protecting longer chains? The data presented in figure 1C would indicate this is the case. In this panel, the TR-TUBE completely protects chains up to 6 moieties, but chains of 7 moieties or longer are clearly

degraded by trypsin (although not completely).

The authors must address this point, e.g. by performing similar experiments with shorter and longer TR-TUBEs (for example 4 UBAs and 8 or 10 UBAs), to clarify if their reported chain lengths and proportions are true. It will be informative to see if chain length distribution varies with the number of domains in the TR-TUBE and if the length of the chains observed with longer TUBEs increases.

FURTHER SPECIFIC COMMENTS requiring re-phrasing:

1) Throughout the manuscript, the authors use the word “architecture” to describe the nature ubiquitin modifications. The authors describe ubiquitin architecture as including “linkage types, Ub PTMs, and chain lengths of endogenous substrates”. To this reviewer, a fourth component dictates overall chain architecture as well: branching, or how the Ub moieties are connected (homotypic, heterotypic, or branched chains). The authors claim in the title that their new method and the data presented in this manuscript “reveals global architecture of ubiquitylation”. This is not true – the method reports on chain length, not architecture.

The authors even mention this in the discussion: “combinations of Ub-ProT, Ub-AQUA/PRM, UbiCRest, and middle-down MS might allow us to investigate the Ub chain architectures of more complex chains, including the positions of linkage branching and PTMs”. Hence, we are quite some way off studying architecture of chains. The authors should rephrase incidents discussing Ub chain architecture.

2) The authors refer to cdc48 as modulating chain length. This is imprecise, as cdc48 itself cannot regulate chain length directly. This is likely done via cdc48-associated Ub ligases and DUBs, as has been proposed in the literature. Hence, while the claim in itself may be correct, the activity to do this is likely in the various associated factors of cdc48. This should be rephrased, especially in the abstract.

3) The authors should be careful about where their method may mislead. A Ub chains which is branched off and sports monoubiquitination events at each Ub molecule would alter result in AQUA/PRM results that the authors do not seem to consider. Also, in the EGFR case, 40% monoUb and 50% K63 chains is inconsistent with eg tetraUb modified EGFR (which should be 75% K63 and 25% unmodified). Here, the TR-TUBE does not seem to have captured monoubiquitinated EGFR (and I would not expect it to, necessarily). The authors should look carefully at the phrasing of some of the results description, and tone down some of the conclusions with more accommodating language.

Minor points:

1. P. 7, ll. 148-149: the authors state “TR-TUBE captured almost all endogenous ubiquitylated proteins other than Ub monomers (Fig. 2a, lanes 1 and 5)”. This statement alludes to a quantitative assessment of the efficiency of the capture by the TR-TUBE. But that cannot be concluded from the data in Fig 2a. They authors should address the quantitative efficiency properly or rephrase this sentence to illustrate that it is a qualitative assessment of the range of Ub modifications that the TR-TUBE captures.
2. P. 9, ll. 201-202: they authors could comment on or discuss the phenotypes observed in the ubp6 and rad23dsk2 strains. This reviewer is particularly puzzled by the apparent decrease in cellular Ub conjugates in the rad23dsk2 strain.
3. The reference to the pdr5 mutant is missing, and the description of this mutant needs to be expanded in the main text.
4. Fig 1a: These are not all the possible permutations of a substrate modified with 4 Ub moieties. The authors completely ignore, as alluded to above, the possibility of branching. An Ub chain with the apparent molecular weight of 32 kDa might be a homotypic tetra-chain, but it might also be a tri-chain with a branch point.
5. Fig 1c, K48 panel, lane 2 (input + trypsin): where has the monoUb band gone? It is present in the other panels. Has this samples been treated longer than the 63 and M1 samples?
6. Fig 3a, PD panel: In the Ub-ProT lanes it looks like there is hardly any Ub chains released in the WT, ubp6, and r23d2 lanes? A longer exposure of that blot would be informative to be able to see that Ub chains released from the substrates in the samples.
7. Fig 4d: The authors should perform AQUA/PRM analysis on the gel slices of the Ub chains released from EGFR, similar to Fig 2 and Fig 3, to get a quantitative assessment of the chain length.
8. Introduction, p3 end of 1st paragraph: Refs 5 and 6 should be cited together with Ref 4 - all three reviews cover the expanded Ub code.
9. A comment should be added about how and where this reagent may be made available to researchers in the field.

Response to the reviewers' comments

We thank the reviewers for their interest in our work and their constructive criticism. We have addressed all comments to the best of our abilities, and have greatly improved the manuscript.

A total of 10 pieces of new data have been added in the revised manuscript (Figs.1c, 1d, 3b, 5a, 5b Supplementary Figs. 2c, 2d, 3, 4a, 4b and 5; Supplementary Table 1). Please find below a point-by-point response addressing each comment.

Reviewers' comments:

Reviewer #1 (Remarks to the Author):

This manuscript describes a method that enables the identification of polyubiquitylation motifs in the yeast proteome using affinity purification and middle down proteomics. They used a previously reported trypsin-resistant tandem ubiquitin-binding entity(ies) (TR-TUBE) to bind modified substrates and prevent the cleavage of the polyubiquitin chains by trypsin (Yoshida et al., Proc Natl Acad Sci U S A. 2015, 112(15):4630-5). The resulting sample is loaded on SDS-PAGE and resolved based on polymeric ubiquitin chain linkages. The SDS-PAGE bands are subjected to in gel digested and the ubiquitylation linkages are quantified using synthetic peptides (AQUA). Using this method, the authors show the importance of Cdc48 in regulating polyubiquitin chain length. The authors also used their method to determine ubiquitin chain linkages found on EGFR upon ligand binding. The method described here is of potential interest to Nature Commun readership as it facilitates the identification of some polyubiquitin motifs. To fully appreciate the originality of this contribution, the authors must also contrast its benefits since this affinity purification approach was previous reported by the same group (Yoshida et al.). Minor comments are highlighted below.

Thank you for your comments, and your reference of Yoshida et al., PNAS 2015. We have added citations for Yoshida et al as suggested, and for Tsuchiya et al., Mol Cell, 2017. While we use a similar technique of TR-TUBE as in the previous papers, we have modified this technique to detect chain lengths in the current manuscript, a strategy that was not reported previously. Thus we believe our current manuscript is original, and provides benefits not present in the previous publications. In the current version, we have modified the text to clarify this point, and also added new data (eg. identification of Npl4 in regulating chain length, Fig. 5) and new controls (eg. use of different length TR-TUBE molecules, Supplemental Fig. 4) to verify the benefits and reliability of our method.

On page 8, the authors mention “A previous proteomics study showed that yeast tryptic peptides are, on average, 8.4 amino acids in length. Thus, if the ubiquitylation sites were structurally hindered (i.e., if trypsin were unable to attack the proximal ubiquitins), the substrate-attached ubiquitin chains should converge to the individual chain sizes.” This phrase should be restructured or removed, as it leads to confusion.

Moreover, the ubiquitin found on the target protein is digested readily by trypsin since there is a commercial antibody that recognized the diglycine motif on ubiquitin modified peptides.

We have removed the confusing sentences.

Figure 1d, Can the authors comment on the occurrence of intact ubiquitin after tryptic digestion?

Trypsin preferentially cleaves ubiquitin at exposed Arg74, digesting Ub chains to monomer size. While trypsin can further digest monomers, monomers have a highly structured, stable conformation that is more resistant to digestion. Our trypsin treatment digests all polyubiquitin chains, but occasionally some monoubiquitin remains after digestion (Fig. 1c and d). We have added a statement explaining this in the figure legend. This monoubiquitin that occasionally remains does not alter measurements of polyubiquitin chain lengths.

Figure 2a. Why there is no ubiquitin monomer in the PD: TR-TUBE lanes but there is free ubiquitin in panel C (middle gel)?

The reviewer asks why our anti-Ub westerns show monomers in Fig. 3c middle panel (formerly Fig. 2c) but not in Fig. 3a (formerly Fig. 2a). The blot in Fig. 3c was used in an experiment where different gel fragments corresponding to different length chains were analyzed for linkage type frequencies. Thus, 5 and 15 fold higher protein amounts were run in gels in Fig. 3c compared to the gel run in Fig 3a. In an experiment added as Supplementary Fig. 3, where we compare oriole stained Ub ladders to Ub ladders detected by western, we observed that our anti-Ub antibody detects Ub monomers with much lower affinity than multimers, and cannot be used to accurately estimate monomer amounts. Thus the 5 fold lower amounts of sample run in Fig. 3a is likely close to the limits of detection for monomers for our antibody. We have added this explanation to the figure legends.

Reviewer #2 (Remarks to the Author):

Tsuchiya et al. report on a novel method termed “ubiquitin chain protection from trypsination” (Ub-ProT) to determine the length of ubiquitin (Ub) chains on client proteins. In Ub-ProT ubiquitylated material is isolated by employing a protein termed TR-TUBE, which encompasses six repeats of a trypsin insensitive variant of the UBQLN1 UBA Ub binding domain. First, the authors demonstrate that TR-TUBE binds in vitro synthesised Ub chains linked via different lysine residues. The bound material is then treated with trypsin. Ub chains are protected from digestion by their interaction with the TR-TUBE whereas other proteins are removed. The authors then determine the size of the Ub chains by gel electrophoresis and analyse their composition by quantitative mass spectrometry. As a proof of concept the authors present data on the composition of Ub chains in yeast cell lysates and on the EGF receptor in EGF-treated HeLa cells.

General Points:

The presented study introduces a simple and potentially powerful method to analyse Ub modifications from total cell lysates. The experiments are of a more than adequate quality and in most cases contain appropriate controls. However, I am not convinced on the general versatility of the assay. An essential prerequisite for Ub-ProT is the robust protection of Ub chains by TR-TUBE from trypsin digestion. However, by comparing the amounts of di-Ub and of a lower molecular weight species that, to my opinion, represents mono-Ub in the TR-TUBE lanes in Figure S2c I get the impression that TR-TUBE shields di-Ub linked via distinct lysines from trypsination to a different degree. For example, K48-linked di-Ub appears to be less sensitive to the protease than K6- or K27-linked Ub (compare the amounts of di-Ub in the input and TR-TUBE lanes). This may be attributed to different binding affinities of these molecules or by diverging accessibility of cleavage sites in the bound state.

The reviewer suggests that different linkage types may be protected to different extents in our method. We agree that there is some variation in protection based on linkage type and have added more accurate data and quantification of this in Supplementary Fig. 2c and 2d. Over 90% of K11, K33, K48, K63, and M1 linkages are protected, while 40 to 60% of K6, K27, K29, and K33 linkages are protected. Thus, it is necessary to keep in mind that our method can underestimate measured frequencies of certain linkage types by ~50%. However, despite this variability, all linkage-types are protected compared to controls digested in the absence of TR-TUBE. Thus our conclusions regarding global analysis of yeast Ub chain lengths, and the effect of Cdc48 and Npl4 on Ub chain length remain convincing. For analysis of K63-linked chains on EGFR, again, our conclusions

regarding modification by 4 and 6 mer Ub chains is valid. Furthermore, as seen in new Fig. 3b, our TR-TUBE construct does not significantly skew the frequencies of different linkage types when compared to the frequencies found in lysate. We have added a section to the Discussion (2nd paragraph on page 10 to page 11), summarizing the limitations of our method, including our results indicating that it may underestimate the prevalence of K6, K27, K29, and K33 linked chains by 50%.

In addition, the authors do not convincingly show that the binding to TR-TUBE fully prevents the access of trypsin to long poly-Ub chains of all linkage types (see also my comments below).

To address the issue brought up by the reviewer, we have added data for K11-linked polyubiquitin chains to Fig. 1c. This shows that Ub chains pulled down by TR-TUBE and protected from trypsin digestion are similar to the chains found in the starting sample. Unfortunately, long chains of other ubiquitinating linkages (K6-, K27-, K29- and K33-linked) are not commercially available, and the specific E2 and E3s needed to generate these chains are also not available. Thus, in total, we have analyzed protection for K48, K63, M1, and K11 chains, K48/K63 chains, and ubiquitin dimers of all linkage types. In addition, we have added data indicating that the length of protected Ub chains does not directly depend on the number of Ub binding domains in TR-TUBE (Supplemental Fig. 4). Combined with the data in Fig. 3b demonstrating that TR-TUBE does not significantly alter linkage type frequency, we believe that our method measures Ub chain lengths with reasonable accuracy. We have added an extensive discussion on the limitations of our method in the Discussion section (pages 10-11).

Another critical issue of the assay is the amount of added protease. In their experiments, the authors determined the ideal trypsin concentration using only K48-linked di-Ub (Fig. S2b). In consequence, the experimental conditions allegedly favour the preservation of certain poly-Ub forms more than others, which complicates the analysis of complex samples.

In all experiments, Ub chains were digested to monomers or further in controls where trypsin was added in the absence of TR-TUBE. Thus, even though we determined trypsin amounts using K48-linked di-Ub, we demonstrate that this amount is sufficient to digest Ub chains of all linkage-types in the absence of TR-TUBE. Since monomeric ubiquitin is more resistant to trypsin than chains, occasionally, ubiquitin monomers were observed after trypsinization in some controls, but this did not affect our analysis of chain length or chain composition. Thus, incomplete digestion is unlikely to be a source of error in our experiments. We have added a time course of trypsin digestion for K48 and K63 linked chains as Supplemental Fig. 3.

As the authors recognize, further problems arise from Ub chains containing heterogeneous linkage types or branching points. In summary, the assay can be individually optimized for the study of a defined poly-Ub modification but is in its current state probably less suitable to quantitatively analyse different poly-Ub species in parallel. Given these concerns and the issues listed below I do not recommend publication of this work in its present form (major revision).

As requested by the reviewer, we have significantly revised our manuscript, adding 10 new pieces of data that both extend our findings (eg. characterizing Cdc48 interacting proteins for Ub chain length regulation, Fig. 5), and define the limits of our method (eg. quantification of the protection of different linkages and analysis of protection of heterogeneous linkages). We find that there can be an approximately 2-fold variation in protection of different linkage types by Ub-ProT (Supplementary Fig. 2d). This variation has the potential of affecting quantitative experiments, and we have modified the text in the Results and Discussion sections to point this out. We also find that heterogeneous K48/K63 linkages can also be protected from trypsinization by Ub-ProT, although again, there is a decrease in protection efficiency (Fig. 1d and Supplementary Fig. 5). Thus, although our method has limitations, by tailoring its use to specific biological questions, we believe this method will be useful in assessing, qualitative, if not quantitative values for Ub chain lengths, a parameter that has previously been notoriously difficult to measure.

Specific points:

Figure 1C: Why is mono-Ub completely digested by trypsin in the K48 sample, while it remains largely untouched in the K63 and M1 experiments?

We chose our trypsin concentration based on complete cleavage of K48-linked diubiquitin to monoubiquitin, and confirmed that this concentration is sufficient to digest Ub-chains in all experiments described in this manuscript. Monoubiquitin can be digested further by trypsin, but is more resistant to digestion due to its compact structure. Thus, in the K63 and M1 experiments, all Ub chains have been digested to monoubiquitin, and most of the monoubiquitin has been further degraded, but due to variability between experiments, some monoubiquitin remains in these lanes. We have added an explanation of this in the legends, and added a time course experiment for trypsinization of unanchored polyubiquitin chains, visualized by Oriole protein staining and immunoblotting with ubiquitin antibody (new Supplementary Fig. 3).

I also don't agree with the author's argumentation here. In the K48 experiment there is a considerable re-arrangement of poly-Ub in the PD lanes. Upon trypsin treatment the amount of high-molecular weight poly-Ub substantially decreases and at the same time there is an increase in the levels of di-Ub. The authors should employ longer K48- and K63-linked poly-Ub chains to ensure quantitative protection of longer poly-Ub chains by TR-TUBE.

Unfortunately we have not been able to obtain, through purchasing or synthesis, K48 chains longer than heptamers or K63 chains longer than decamers. In place of this, we have obtained and added data from long K11 chains and added this data to Fig. 1c. K11 chains subjected to Ub-ProT are indistinguishable from input chains. This result is similar to previous results we obtained for long M1 chains. For long M1 chains, we have added data indicating that the length of protected chains does not depend on the number of Ub binding domains present on TR-TUBE, suggesting that TR-TUBE binds in tandem to protect long chains (Supplemental Fig. 4). This new data, combined with previous data indicating that Ub-ProT fully protects dimeric K48 and K63 chains (Supplemental Fig 2), and with data indicating that TR-TUBE pull downs do not significantly alter linkage frequencies in yeast lysates (Fig. 3b), suggests that Ub-ProT can be used to reasonably estimate Ub chain lengths. As described above, we address limitations to our method in the Discussion section.

Figure 3: Cdc48 mutants accumulate roughly six fold higher levels of poly-ubiquitylated material than the other yeast strains as assessed by Ub-AQUA/MS analysis (Fig. 3b). Thus, the increased length of Ub chains in the cdc48-3 samples (Fig. 3d) may be explained by insufficient protease capacity, given that a portion of poly-Ub is still sensitive to trypsin even in presence of TR-TUBE. Does the analysis of Ub chain length yield varying results, when trypsin concentration is changed (e.g. normalized to amount of Ub chains)?

The amount of trypsin added to reactions is normalized to total protein amounts rather than amounts of Ub chains. We do not believe our results in Fig. 4 (formerly Fig. 3) are due to insufficient protease capacity since the same amount of trypsin completely digests Ub chains in lysates in the absence of TR-TUBE (Fig. 4a).

The authors sometimes tend to over-interpret their results without considering alternative explanations. For example, the authors state that "Cdc34 and Rsp5 modified themselves with K48- and K63-linked chains, respectively up to 10-mer length" (page 7 lines 134-136). This statement is not fully correct. The authors should consider that longer chains may have not been completely

covered by TR-TUBEs and thus have been partially degraded by trypsin. A closer view on Fig. 1c suggests that high molecular weight Ub chains may indeed not be fully protected from trypsin by TR-TUBE. Thus, a more careful phrasing is required.

We agree with the reviewer regarding the limitations of our method and have added a paragraph in the Discussion section addressing limitations. This paragraph discusses possible skewing of linkage type frequencies due to preferential protection and possibilities regarding measured chain lengths.

The authors also show that TR-TUBE does not bind mono-Ub but still they find a considerable amount of this species in their samples after trypsin treatment. According to their idea, this mono-Ub is derived from proteins containing multiple mono-Ub modifications. However, mono-Ub may also be generated by the trypsination of Ub chains that were not fully protected by TR-TUBEs. In fact, there are small amounts of peptides indicative for K11- and K63- (and traces of K48-) linked Ub found in fractions that correspond to mono-Ub (Fig. 2e & 3d). The authors agree that the binding of TR-TUBE to longer or branched Ub chains has not been systematically tested, and hence some mono-Ub may originate from partial decomposition of such structures.

As the reviewer points out, we did find traces of K11 and K63 linked Ub in Ub monomer regions of the gels. However, for both linkages, the amount found in the monomeric area was less than 5% of the total (please see Dataset 3 in which we summarize the absolute amount of each Ub-PRM data point). For other linkage types, the amounts found in the monomeric fraction were much less. Thus, although there is some minor contamination from degradation of longer or branched chains, our data suggest that the majority of monoubiquitin products are derived from multiple-mono ubiquitylation of substrate proteins. Comments on this have been included in the Discussion.

The authors also find that inactivation of Cdc48 increases the average length of some poly-Ub species. Their interpretation is that the segregation of substrate-ligase complexes by this AAA-ATPase terminates the ubiquitylation reaction and thereby restricts chain length. However, Cdc48 is known to associate with various Ub elongating and de-ubiquitylating enzymes, which were shown to shape the Ub landscape on substrates. Defects in Cdc48 function may affect the activity of these Ub modifying enzymes and thereby cause changes in the overall composition of poly-Ub chains.

As pointed by the reviewer, ubiquitin chains can be remodeled on Cdc48 complexes by multiple cofactors, including Ufd1, Npl4, Shp1, and Ufd3, the deubiquitinase Otu1, and E4 Ub elongation factor Ufd2. To respond the reviewer's comment, we investigated the role of these cofactors in

regulating ubiquitin chain lengths (Fig. 5). Ub-ProT revealed that Ub chains were significantly elongated in the *npl4-1* mutant, which is defective in the extraction of ubiquitylated membrane proteins (Hitchcock et al., 2003) (Fig. 5a). For other cofactors, Ub chains were slightly elongated in *shp1Δ* cells, but unaffected in *otu1Δ*, and *ufd2Δ*, and *ufd3Δ* cells (Fig. 5b). We previously found that the ability for Npl4 to bind Ub is completely abolished in the *npl4-1* mutant (Tsuchiya et al., Mol Cell, 2017). Thus, we conclude that the recognition and segregation of ubiquitylated substrates by the Cdc48 complex (in particular Cdc48 and Npl4) is important in controlling ubiquitin chain lengths in cells. We appreciate the reviewer's insightful advice, which have strengthened our data.

Minor issues:

There are two types of asterisks in Fig. S1d that should be explained in the legend.

We have added the explanation for the two asterisks.

Reviewer #3 (Remarks to the Author):

Review of Tsuchiya et al. – A method for determining ubiquitin chain length and global architecture of protein ubiquitylation in cells. Nature Communications 2017

GENERAL COMMENTS:

In this manuscript, the authors present a new and quite ingenious method for determining Ub chain length on substrates in vitro and in vivo. This method uses Ub chain protection using a Trypsin-resistant TUBE and is termed Ub-ProT. After pulldown, tryptic digest, the power of mass-spectrometry (AQUA/PRM analyses) and careful gel-based analysis is used to reveal Ub chain length samples.

The method and idea are excellent and a big advance for the Ub field as it provides the field of ubiquitin signaling research with a new and potentially useful and valuable tool to understand the size and chain type of Ub modifications on cellular substrates, a task that to this day has remained difficult. The method will lead to a better understanding of endogenous, ubiquitin-modified substrates.

The manuscript is well written and the data is clear. With addressing the sole experimental concern for the study and some careful rephrasing to avoid misunderstandings and not oversell the work, I would support publication.

Below is a list of suggested improvements of the manuscript.

MAIN COMMENT requiring new experiments:

The TR-TUBE used throughout this study consists of 6 UBA domains. Although the authors observe longer chains, especially in the cdc48-3 mutant strain, they observe a peak in chain length on substrates of ~6 Ub moieties in both unperturbed cells in figure 2 and in EGF-stimulated cells in figure 4. This reviewer is wondering if these observed chain lengths are true or whether they are a consequence of the capacity of the TR-TUBE to bind 6 Ub moieties, i.e., are the authors systematically underestimating the true length of the chains and the proportion of longer chains due to the fact that the TR-TUBE might not be proficient in protecting longer chains? The data presented in figure 1C would indicate this is the case. In this panel, the TR-TUBE completely protects chains up to 6 moieties, but chains of 7 moieties or longer are clearly degraded by trypsin (although not completely).

The authors must address this point, e.g. by performing similar experiments with shorter and longer TR-TUBEs (for example 4 UBAs and 8 or 10 UBAs), to clarify if their reported chain lengths and proportions are true. It will be informative to see if chain length distribution varies with the number of domains in the TR-TUBE and if the length of the chains observed with longer TUBEs increases.

We appreciate the reviewer's advice. As requested, we performed Ub-ProT on free linear ubiquitin chains using 4×, 6× and 8×UBAs (Supplementary Fig. 4). All TUBE constructs protected longer linear polyubiquitin chains from trypsinization, and there were no apparent differences in protection between constructs. Furthermore, chains Ub-ProT showed no apparent differences in sizes and intensities compared to input chains. Thus our results are unlikely to be an artifact of individual TR-TUBE capacities.

FURTHER SPECIFIC COMMENTS requiring re-phrasing:

1) Throughout the manuscript, the authors use the word "architecture" to describe the nature ubiquitin modifications. The authors describe ubiquitin architecture as including "linkage types, Ub PTMs, and chain lengths of endogenous substrates". To this reviewer, a fourth component dictates overall chain architecture as well: branching, or how the Ub moieties are connected (homotypic, heterotypic, or branched chains). The authors claim in the title that their new method and the data

presented in this manuscript “reveals global architecture of ubiquitylation”. This is not true – the method reports on chain length, not architecture.

The authors even mention this the discussion: “combinations of Ub-ProT, Ub-AQUA/PRM, UbiCRest, and middle-down MS might allow us to investigate the Ub chain architectures of more complex chains, including the positions of linkage branching and PTMs”. Hence, we are quite some way off studying architecture of chains. The authors should rephrase incidents discussing Ub chain architecture.

We thank the reviewer for pointing this out. We have changed the title of the manuscript to “Ub-ProT reveals global length and composition of protein ubiquitylation in cells”, and have been more careful in our wording throughout the text.

2) The authors refer to cdc48 as modulating chain length. This is imprecise, as cdc48 itself cannot regulate chain length directly. This is likely done via cdc48-associated Ub ligases and DUBs, as has been proposed in the literature. Hence, while the claim in itself may be correct, the activity to do this is likely in the various associated factors of cdc48. This should be rephrased, especially in the abstract.

In response the reviewer comment, we performed Ub-ProT using Cdc48 cofactor mutants, and found that Ub chains were significantly elongated in the *npl4-1* mutant, which is defective in the extraction of ubiquitylated membrane proteins (Hitchcock et al., 2003) (Fig. 5a). For other cofactors, Ub chains were slightly elongated in *shp1Δ* cells, but unaffected in *otu1Δ*, and *ufd2Δ*, and *ufd3Δ* cells (Fig. 5b). Thus, we propose that the segregation activity of the Cdc48 complex, and in particular Cdc48 and Npl4, results in inhibition of Ub chain elongation.

3) The authors should be careful about where their method may mislead. A Ub chains which is branched off and sports monoubiquitination events at each Ub molecule would alter result in AQUA/PRM results that the authors do not seem to consider. Also, in the EGFR case, 40% monoUb and 50% K63 chains is inconsistent with eg tetraUb modified EGFR (which should be 75% K63 and 25% unmodified). Here, the TR-TUBE does not seem to have captured monoubiquitinated EGFR (and I would not expect it to, necessarily). The authors should look carefully at the phrasing of some of the results description, and tone down some of the conclusions with more accommodating language.

As the reviewer points out, if EGF were only modified with K63 linked tetra Ub chains, we would

expect linkage analysis to yield 75% K63 and 25% mono/endcap. Since we did not find this ratio (Fig. 6c), we proposed the model in Fig. 6f where EGF is also modified with monoubiquitin. However, we agree that branched terminal Ub that is not protected by TR-TUBE may also explain our results. We have modified our text accordingly, and included a paragraph in the Discussion section addressing the limitations of our method.

Minor points:

1. P. 7, ll. 148-149: the authors state “TR-TUBE captured almost all endogenous ubiquitylated proteins other than Ub monomers (Fig. 2a, lanes 1 and 5)”. This statement alludes to a quantitative assessment of the efficiency of the capture by the TR-TUBE. But that cannot be concluded from the data in Fig 2a. They authors should address the quantitative efficiency properly or rephrase this sentence to illustrate that it is a qualitative assessment of the range of Ub modifications that the TR-TUBE captures.

We have removed the quantitative phrase, “almost all,” and replaced the sentence with, “Immunoblotting with anti-Ub antibody revealed that TR-TUBE was unable to pull down Ub monomers, but otherwise captured endogenous ubiquitylated proteins efficiently (Fig. 3a, lanes 1 and 5).” For quantitative binding efficiency, we measured the ability of Ub-ProT to protect dimeric Ub of all linkage types in Supplemental Fig. 2 and added the results to the Result and Discussion.

2. P. 9, ll. 201-202: they authors could comment on or discuss the phenotypes observed in the ubp6 and rad23dsk2 strains. This reviewer is particularly puzzled by the apparent decrease in cellular Ub conjugates in the rad23dsk2 strain.

Our result is consistent with results from a previous study demonstrating that Rad23, Dsk2, and Rpn10 protect ubiquitin chains from deubiquitylating enzymes (Hartmann-Petersen, FEBS Letters, 2003). While these factors contribute to proteasomal degradation by protecting and delivering ubiquitylated substrates, our results suggest that in the absence of these factors, ubiquitylation is reduced by deubiquitylating enzyme activity. We have added a statement to this effect in the Results section, page 7, bottom.

3. The reference to the pdr5 mutant is missing, and the description of this mutant needs to be expanded in the main text.

We have inserted a reference for Fleming et al., 2002, and expanded the text to explain that the *pdr5* mutation increases sensitivity to the proteasome inhibitor MG132.

4. Fig 1a: These are not all the possible permutations of a substrate modified with 4 Ub moieties. The authors completely ignore, as alluded to above, the possibility of branching. An Ub chain with the apparent molecular weight of 32 kDa might be a homotypic tetra-chain, but it might also be a tri-chain with a branch point.

As pointed out by the reviewer, we have ignored both linkage types and branching in Fig. 1a. We have modified the text in the Results section to indicate that the five different topologies shown in the figure ignore linkage type and branching. We have also modified the figure legend to point this out.

5. Fig 1c, K48 panel, lane 2 (input + trypsin): where has the monoUb band gone? It is present in the other panels. Has this samples been treated longer than the 63 and M1 samples?

Monoubiquitin is more resistant to trypsin digestion compared to polyubiquitin, due to its compact structure. We titrated trypsin amounts to digest Ub chains, so small amounts of monoubiquitin sometimes remain. For digestion of K63 and M1 linked chains, we believe that all of the chains were digested to mono ubiquitin, and most of the mono ubiquitin was further digested. However, a small amount remains which resembles the amount of mono ubiquitin found in the input lanes. For the K48 chain, we believe mono ubiquitin was completely digested due to slight experimental variation. We did not treat this sample any differently from the others. We have inserted a sentence to explain this in the legend.

6. Fig 3a, PD panel: In the Ub-ProT lanes it looks like there is hardly any Ub chains released in the WT, ubp6, and r23d2 lanes? A longer exposure of that blot would be informative to be able to see that Ub chains released from the substrates in the samples.

As suggested, we have added a longer exposure of the Ub-ProT lanes of Fig. 3a (currently Fig. 4a), which demonstrates that Ub-ProT does protect Ub chains in WT, ubp6, and r23d2 lysates.

7. Fig 4d: The authors should perform AQUA/PRM analysis on the gel slices of the Ub chains released from EGFR, similar to Fig 2 and Fig 3, to get a quantitative assessment of the chain length.

The amount of ubiquitin chains that we were able to isolate from EGFR was very low, so that the amount of Ub in each gel sliced fraction would be near the lower limit of quantitation by Ub-AQUA/PRM (Supplementary Fig.7). Therefore, we instead used the ubiquitin chain restriction (UbiCRest) assay developed by Komander et al., it is difficult to get more information from this samples. Instead of the quantitative assessment of the chain length, we performed UbiCRest assay to consider to ubiquitin chain topologies.

8. Introduction, p3 end of 1st paragraph: Refs 5 and 6 should be cited together with Ref 4 - all three reviews cover the expanded Ub code.

We have grouped references 4, 5 and 6 to cite the expanded Ub code.

9. A comment should be added about how and where this reagent may be made available to researchers in the field.

We will add the reagent to Addgene and have included a sentence to that effect in the Materials and Methods.

Reviewers' Comments:

Reviewer #2 (Remarks to the Author):

In the revised version of their manuscript, Tsuchiya et al. included new experiments to address my initial concerns. They now provide stronger evidence that the employed TUBE also protects long ubiquitin chains from trypsination by comparing the protease sensitivity of ubiquitin chains associated with TUBEs containing different numbers of ubiquitin binding domains. They also did some re-phrasing of the text to more accurately describe and interpret their experimental data and included a section discussing the technical limitations of their method. By investigating a yeast strain defective for the function of the Cdc48 partner protein Npl4, they added an accessory experiment to strengthen their view on the impact of Cdc48 on ubiquitin chain re-modeling. In summary, the authors did a good job in answering my questions. I do not fully agree with their interpretation of the results shown in Figure 3b. The authors state that “Linkage composition were similar between lysates and pulled-down samples.” (page 6) and “... this two fold variation did not significantly affect the results of our proof of principle experiments” (page 10). There is only half the relative amount of K63-linked ubiquitin in the Ub-ProT sample compared to the lysate and the TR-TUBE sample detectable, whereas roughly twice the relative amount of K29-linked molecules accumulates here. To my opinion, these changes can be regarded as “significant”. This is a minor issue and should be addressed by a more careful phrasing of the text. Other than that, I can now recommend publishing of this work in “Nature Communications”.

Reviewer #3 (Remarks to the Author):

The authors have addressed all my points and I remain highly enthusiastic.

Response to the reviewers' comments

We again thank all the reviewers for the positive responses and constructive criticisms on our study.

Reviewers' comments:

Reviewer #2 (Remarks to the Author):

In the revised version of their manuscript, Tsuchiya et al. included new experiments to address my initial concerns. They now provide stronger evidence that the employed TUBE also protects long ubiquitin chains from trypsination by comparing the protease sensitivity of ubiquitin chains associated with TUBEs containing different numbers of ubiquitin binding domains. They also did some re-phrasing of the text to more accurately describe and interpret their experimental data and included a section discussing the technical limitations of their method. By investigating a yeast strain defective for the function of the Cdc48 partner protein Npl4, they added an accessory experiment to strengthen their view on the impact of Cdc48 on ubiquitin chain re-modeling. In summary, the authors did a good job in answering my questions. I do not fully agree with their interpretation of the results shown in Figure 3b. The authors state that "Linkage composition were similar between lysates and pulled-down samples." (page 6) and "... this two fold variation did not significantly affect the results of our proof of principle experiments" (page 10). There is only half the relative amount of K63-linked ubiquitin in the Ub-ProT sample compared to the lysate and the TR-TUBE sample detectable, whereas roughly twice the relative amount of K29-linked molecules accumulates here. To my opinion, these changes can be regarded as "significant". This is a minor issue and should be addressed by a more careful phrasing of the text. Other than that, I can now recommend publishing of this work in "Nature Communications".

We appreciate the reviewer's favorable and constructive comments. With respect to the interpretation of Figure 3b, we admit that the quantitative values of Ub-ProT inevitably varied due to gel handling etc (Figure 3d). For this reason, we deleted the sentence "Although this two fold variation did not significantly affect the results of our proof of principle experiments (e.g. Fig. 3b, compared to input with Ub-ProT)" and revised to "Therefore, our method may slightly underestimate the occurrence of certain linkage-types." in Discussion (page 10).

Reviewer #3 (Remarks to the Author):

The authors have addressed all my points and I remain highly enthusiastic..

We appreciate the favorable comment of the reviewer.